# An electric molecular motor

Long Zhang[1✉], Yunyan Qiu[1], Wei-Guang Liu[2], Hongliang Chen[1,3,4], Dengke Shen[1,5], Bo Song[1], Kang Cai[1,6], Huang Wu[1], Yang Jiao[1], Yuanning Feng[1], James S. W. Seale[1], Cristian Pezzato[1,7,8], Jia Tian[9], Yu Tan[1,10], Xiao-Yang Chen[1], Qing-Hui Guo[1,3,4], Charlotte L. Stern[1], Douglas Philp[1,11], R. Dean Astumian[12✉], William A. Goddard III[2✉] & J. Fraser Stoddart[1,3,4,13✉]

Macroscopic electric motors continue to have a large impact on almost every aspect of modern society. Consequently, the effort towards developing molecular motors[1–3] that can be driven by electricity could not be more timely. Here we describe an electric molecular motor based on a [3]catenane[4,5], in which two cyclobis(paraquat-*p*-phenylene)[6] (CBPQT[4+]) rings are powered by electricity in solution to circumrotate unidirectionally around a 50-membered loop. The constitution of the loop ensures that both rings undergo highly (85%) unidirectional movement under the guidance of a flashing energy ratchet[7,8], whereas the interactions between the two rings give rise to a two-dimensional potential energy surface (PES) similar to that shown by $F_OF_1$ ATP synthase[9]. The unidirectionality is powered by an oscillating[10] voltage[11,12] or external modulation of the redox potential[13]. Initially, we focused our attention on the homologous [2]catenane, only to find that the kinetic asymmetry was insufficient to support unidirectional movement of the sole ring. Accordingly, we incorporated a second CBPQT[4+] ring to provide further symmetry breaking by interactions between the two mobile rings. This demonstration of electrically driven continual circumrotatory motion of two rings around a loop in a [3]catenane is free from the production of waste products and represents an important step towards surface-bound[14] electric molecular motors.

During the past 40 years, the design and synthesis[15,16] of artificial molecular machines have fostered the promise[17–19] of a technological revolution similar in magnitude to that arising from the development of macroscopic motors. To convert energy from an external source into unidirectional movement on a molecular scale[20,21], several artificial molecular machines, including[3,22,23] rotaxane-based linear motors[5,8,24–27] and catenane-based rotary motors[4,5,28,29], have been designed, synthesized and shown to work in the presence of light[4,24,30–32] and chemical fuels[5,25,29,33,34]. Although it has been demonstrated[35–37] that single-molecule motors can be powered using tunnelling currents under ultrahigh vacuum on surfaces, examples of electrically powered catenane rotary motors operating in solution are not known to the best of our knowledge. Here we report the design, synthesis and operation of a redox-driven rotary motor based on a [3]catenane in which two rings can be powered by electricity to rotate unidirectionally around a loop.

The [3]catenane molecular motor **[3]CMM** comprises (Fig. 1a) two CBPQT[4+] (ref. [6]) rings encircling a 50-membered loop. A bis(4-methylenephenyl)methane (BPM) unit separates two viologen (V[2+]) units, which are preordained, on reduction to their V[+•] reduced radical cationic states, to serve as recognition sites[38] for reduced CBPQT[2(+•)] rings. The remainder of the loop is composed of a chain containing 11

methylene groups and one oxygen atom, intercepted along its length by an isopropylphenylene (IPP) steric barrier, a triazole (T) ring, which is generated during the final ring closure of the loop to give the [3]catenane, and a 2,6-dimethylpyridinium (PY[+]) Coulombic barrier. As a consequence of the redox properties of the V[2+/+•] units in the loop and the two CBPQT[4+/2(+•)] rings, the oxidized state **[3]CMM[13+]** can be converted (Fig. 1b) into a reduced state **[3]CMM[7+6•]**, accompanied by movement of the two CBPQT[2(+•)] rings, so as to encircle the V[+•] units in the loop by virtue of radical-pairing interactions[39]. This design received its inspiration from previous investigations[12,25,40,41] on the redox-driven rotaxane-based molecular pumps, in which a pumping cassette[26,42,43], comprising PY[+]–V[2+]–IPP, serves as an active one-way gate to transport a CBPQT[4+] ring from the bulk solution onto a collecting chain after every redox cycle. In comparison with these linear rotaxane-based pumps, catenane-based motors are expected to undergo continuous unidirectional circumrotation of one ring about the other, as long as there is a power source. The obvious design in which the two ends of a rotaxane-based pump are linked to form a [2]catenane gives rise to a constitution in which there is no probe of unidirectionality. Consequently, we developed the loop and the [3]catenane to investigate the interactions between and the unidirectionality of the two mobile rings.

[1]Department of Chemistry, Northwestern University, Evanston, IL, USA. [2]Materials and Process Simulation Center, California Institute of Technology, Pasadena, CA, USA. [3]Stoddart Institute of Molecular Science, Department of Chemistry, Zhejiang University, Hangzhou, China. [4]ZJU-Hangzhou Global Scientific and Technological Innovation Center, Hangzhou, China. [5]Institutes of Physical Science and Information Technology, Anhui University, Hefei, China. [6]Department of Chemistry, Nankai University, Tianjin, China. [7]Institut des Sciences et Ingénierie Chimiques, École Polytechnique Fédérale de Lausanne (EPFL), Lausanne, Switzerland. [8]Department of Chemical Sciences, University of Padova, Padova, Italy. [9]Key Laboratory of Synthetic and Self-Assembly Chemistry for Organic Functional Molecules, Shanghai Institute of Organic Chemistry, Chinese Academy of Sciences, Shanghai, China. [10]School of Chemical Engineering and Technology, Sun Yat-sen University, Zhuhai, China. [11]School of Chemistry, University of St Andrews, North Haugh, St Andrews, UK. [12]Department of Physics and Astronomy, University of Maine, Orono, ME, USA. [13]School of Chemistry, University of New South Wales, Sydney, New South Wales, Australia. ✉e-mail: long.zhang@northwestern.edu; astumian@maine.edu; wag@caltech.edu; stoddart@northwestern.edu

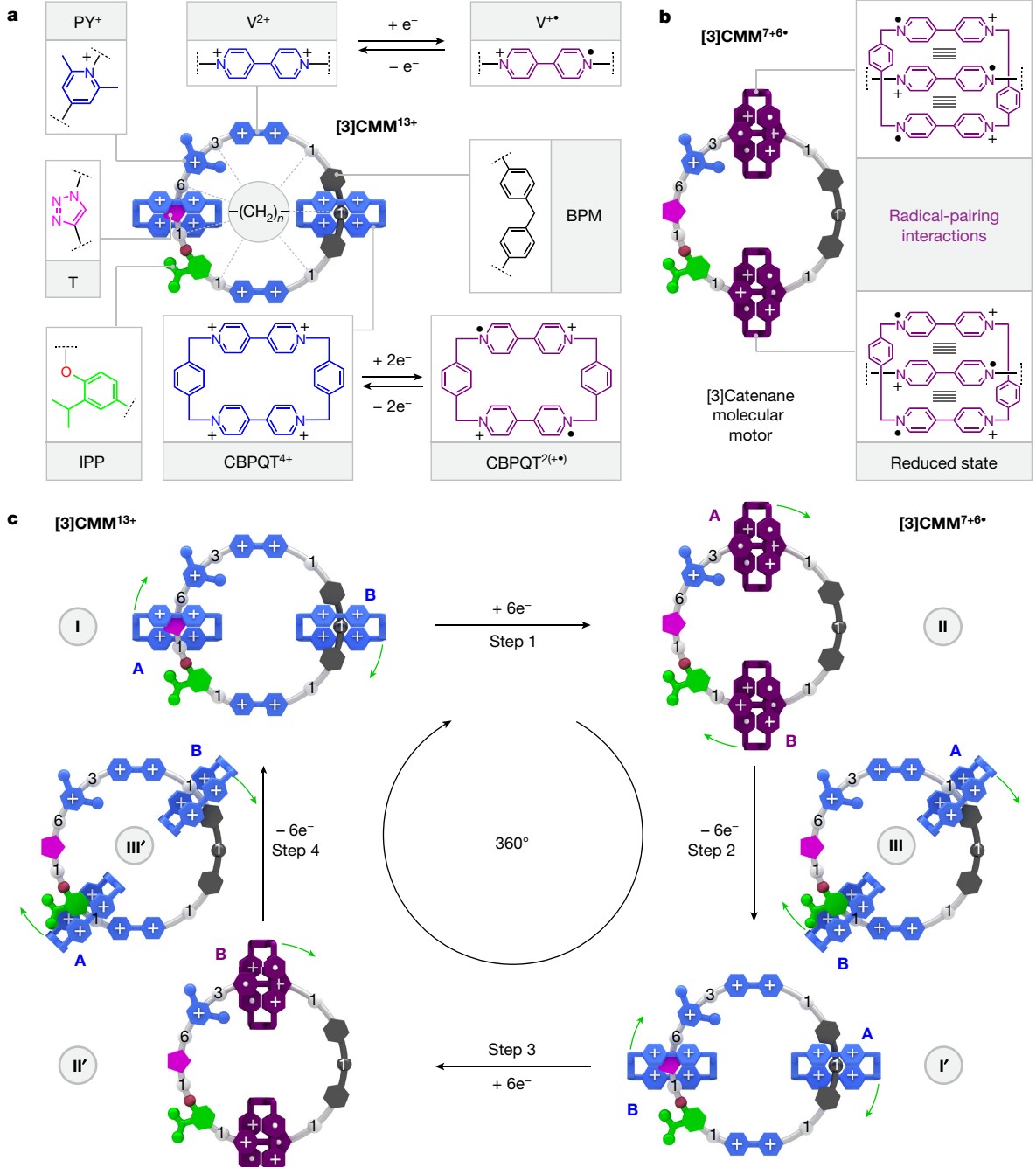

**Fig. 1 | Design and working mechanism of the [3]catenane molecular motor [3]CMM. a**, Graphical representations with key structural fragments for the oxidized state of the [3]catenane molecular motor **[3]CMM¹³⁺**. The cyclobis (paraquat-*p*-phenylene) rings, the bisradical dicationic states of cyclobis (paraquat-*p*-phenylene), the viologens, the radical cationic states of the viologens, the bis(4-methylenephenyl)methane, the isopropylphenylene, the triazole and the 2,6-dimethypyridinium units are labelled as CBPQT⁴⁺, CBPQT²⁽⁺•⁾, V²⁺, V⁺•, BPM, IPP, T and PY⁺, respectively. **b**, Graphical representations for the reduced state of the [3]catenane molecular motor **[3]CMM⁷⁺⁶•** with key superstructural formulas showing the radical-pairing interactions between the CBPQT²⁽⁺•⁾ rings and the V⁺• units. Positive charges are balanced by PF₆⁻ counterions, which are omitted for the sake of clarity. **c**, The redox operation of **[3]CMM¹³⁺/⁷⁺⁶•** demonstrating the unidirectional rotary motion of the two

CBPQT⁴⁺/CBPQT²⁺• rings. In state I, [CBPQT-A]⁴⁺ and [CBPQT-B]⁴⁺ are positioned around the T and BPM units, respectively. Reduction of the V²⁺ units and the CBPQT⁴⁺ rings by the injection (step 1) of six electrons in total triggers both rings to undergo a clockwise rotation, leading to the formation (state II) of the reduced state **[3]CMM⁷⁺⁶•**. Subsequent oxidation by the removal (step 2) of six electrons restores the Coulombic repulsion between the two rings and the loop, obliging [CBPQT-B]⁴⁺ to thread over (state III) the steric barrier (IPP) under thermal activation and eventually encircle T, whereas [CBPQT-A]⁴⁺ finds itself threaded around BPM, thus completing a 180° positional exchange between the two rings shown in state I′. A second redox cycle (steps 3 and 4) resets the system back to state I after the accomplishment of another 180° positional exchange between the two rings.

Initially, we studied (Supplementary Scheme 5) the homologous [2]catenane **[2]C⁹⁺**, in which only one CBPQT⁴⁺ ring encircles the BPM unit on the loop. No directional motion was observed (Supplementary

Fig. 35) in the [2]catenane. Instead, the single CBPQT⁴⁺/²⁽⁺•⁾ ring switches between one of the V²⁺/⁺• units and the BPM unit during a redox cycle. To gain further insight into the switching behaviour of the [2]catenane,

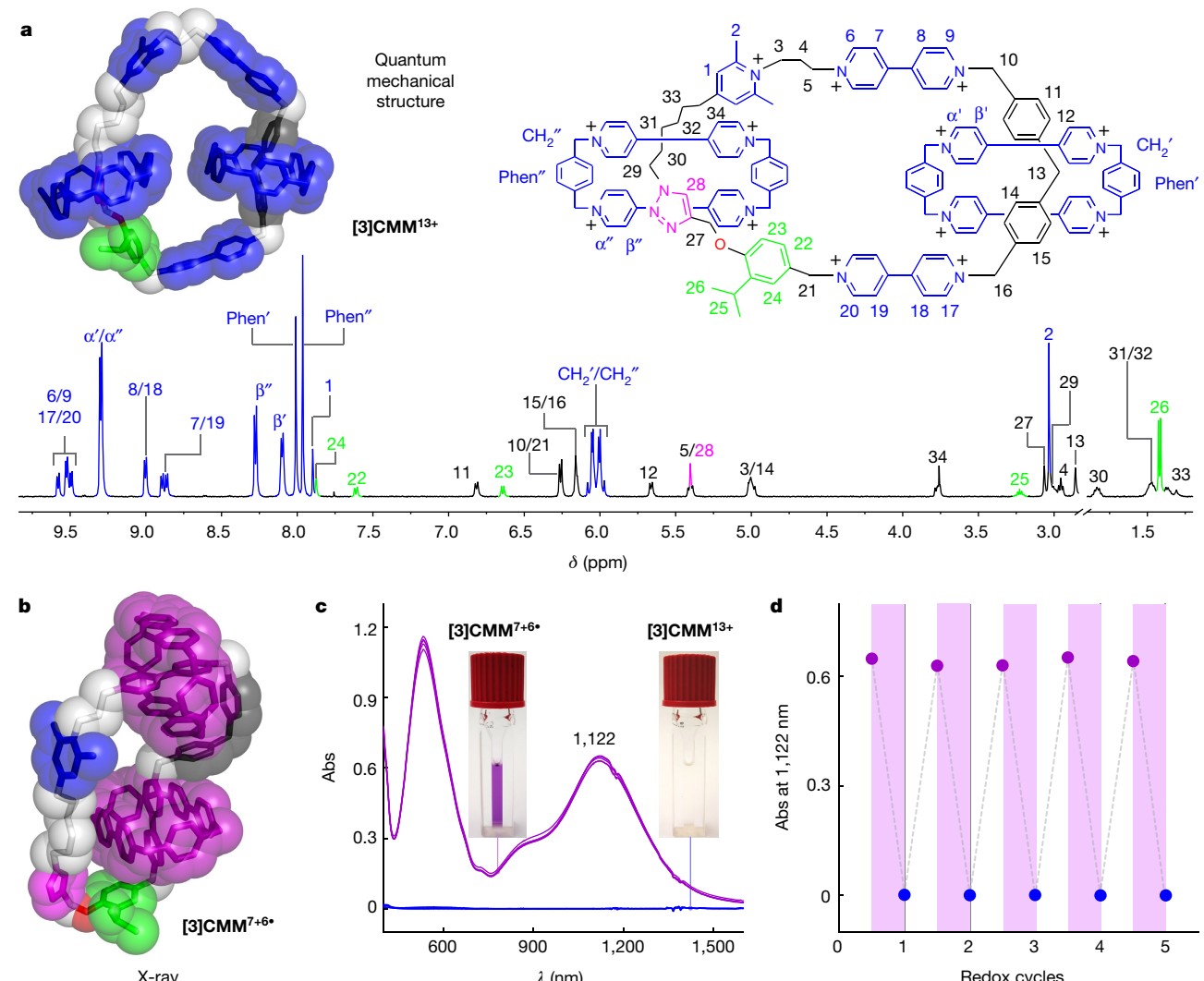

**Fig. 2 | Characterization of the redox state and electrically driven operation of the [3]catenane molecular motor [3]CMM. a**, Structural formula for the oxidized state **[3]CMM¹³⁺** with an optimized quantum mechanical model structure (M06-2X/6-31G* basis set, in tubular with superimposed space-filling representation) and the ¹H NMR spectrum (500 MHz, CD₃COCD₃, 298 K), in which all the proton assignments are labelled. **b**, X-ray single-crystal structure of the reduced state **[3]CMM⁷⁺⁶•** depicted by tubular with superimposed space-filling representations. Solvent molecules, counterions and hydrogen atoms are omitted for the sake of clarity. **c**, Vis/NIR spectra of the reduced state

**[3]CMM⁷⁺⁶•** (purple) and the oxidized state **[3]CMM¹³⁺** (blue) during the electrically driven operation of the molecular motor. Conditions: **[3]CMM** (30 μM), MeCN solution with TBAPF₆ (0.1 M) as the supporting electrolyte, reduction potential −0.5 V (versus Ag/AgCl) for 10 min, oxidation potential +0.7 V (versus Ag/AgCl) for 15 min. Insets are photographs of the two solutions in the oxidized (colourless) and reduced (purple) states. **d**, Absorption intensities of **[3]CMM⁷⁺⁶•** (purple) and **[3]CMM¹³⁺** (blue) at 1,122 nm, showing the reversible switching between the two redox states during each cycle. Abs, absorption; ppm, parts per million.

quantum mechanical calculations were undertaken (Supplementary Information Section 6.1). Several features were shown (Extended Data Figs. 1 and 2) in the calculated PESs of the oxidized and reduced forms of the [2]catenane. (1) The energy of the electrostatic barriers arising from the PY⁺ and V²⁺/⁺• units, as well as the interaction energies between the CBPQT⁴⁺/²⁽⁺•⁾ ring and the V²⁺/⁺• units, can be modulated simultaneously by switching between the oxidized and reduced states of the [2]catenane. (2) The height of the steric barrier imposed by the IPP unit is independent of the redox state and is large enough (ΔE > 25 kcal mol⁻¹) to preclude the passage of the CBPQT⁴⁺/²⁽⁺•⁾ ring over the IPP unit. (3) An energy well, close to the centre of the BPM unit, traps the CBPQT⁴⁺ ring under oxidizing conditions, thereby preventing directional movement on relevant timescales.

These results motivated our investigation of the [3]catenane. When two CBPQT⁴⁺ rings encircle the loop under oxidizing conditions, one

ring occupies the well provided by the BPM unit, thus requiring the passage of the other ring over the IPP unit. This combination of two mechanically interlocked[44] CBPQT⁴⁺/²⁽⁺•⁾ rings, a switchable barrier, associated with the PY⁺ unit, two switchable viologen (V²⁺/⁺•) recognition sites and a steric barrier (IPP) is key to inducing unidirectional motion (Fig. 1c) of the two rings with respect to the loop. Let us consider the mechanism of operation (Fig. 1c). To aid the discussion, the CBPQT⁴⁺ rings are given the descriptors [CBPQT-A]⁴⁺ and [CBPQT-B]⁴⁺ (Fig. 1c, I). At the outset, on account of repulsive Coulombic forces, [CBPQT-A]⁴⁺ resides around the T unit, whereas [CBPQT-B]⁴⁺ encircles the BPM unit. On reduction, six electrons are injected (Fig. 1c, step 1) into the [3]catenane, resulting in both CBPQT⁴⁺ rings being reduced to CBPQT²⁽⁺•⁾ and the V²⁺ units to V⁺•. The Coulombic interactions are decreased, allowing the attractive radical–radical interactions to dominate (Fig. 1c, II). As indicated (Extended Data Fig. 3) by the PESs of the reduced

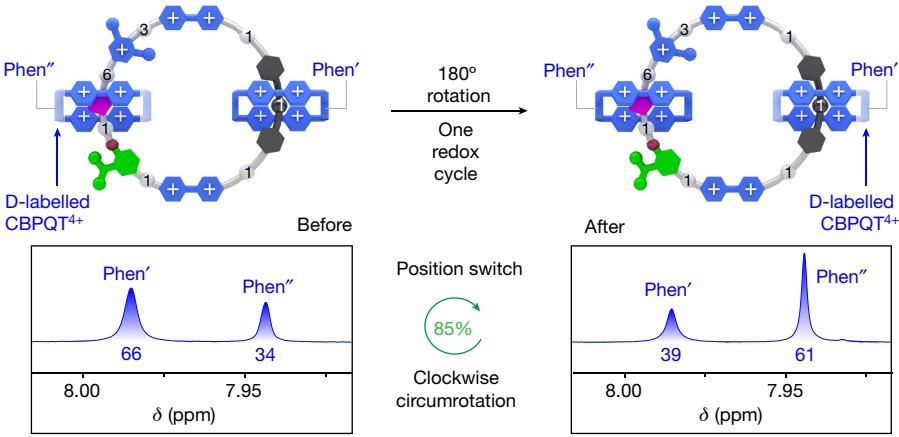

**Fig. 3 | Measurement of the unidirectionality.** Top: graphical representation of the positional exchange of the deuterium-labelled CBPQT⁴⁺ and CBPQT⁴⁺ rings on the loop after one redox cycle. Bottom: partial ¹H NMR (600 MHz, CD₃COCD₃, 298 K) spectra of [Dₙ]-**[3]CMM**·13PF₆ with proton assignments before (left) and after (right) one electrically driven redox cycle. Numbers under peaks indicate relative integrals. ppm, parts per million.

[3]catenane, the relative energy barrier for a CBPQT²⁽⁺⁾ ring to traverse the PY⁺ unit is much lower than the barrier for the same ring to traverse the IPP unit. It follows that [CBPQT-A]²⁽⁺⁾ passes over the PY⁺ unit to thread on to V⁺ in a clockwise direction and [CBPQT-B]²⁽⁺⁾ moves concomitantly to occupy the V⁺ unit that neighbours the IPP unit. In this overall reduced state, the stable **[3]CMM⁷⁺⁶** form is the one in which both CBPQT²⁽⁺⁾ rings encircle[38] the V⁺ binding sites. Subsequent oxidation (Fig. 1c, step 2) leads (Extended Data Fig. 4) to a biased Brownian motion[7,45] (Fig. 1c, III) to install the [CBPQT-A]⁴⁺ ring on the BPM unit and the [CBPQT-B]⁴⁺ ring on the T unit. Overall, one redox cycle triggers the positional exchange (Fig. 1c, I′) between the two CBPQT⁴⁺ rings and completes a 180° unidirectional rotation. A subsequent redox cycle (Fig. 1c, steps 3 and 4) brings the rings back to their initial starting positions and finishes a full 360° clockwise circumrotation around the loop. Steps 3 and 4 are identical to steps 1 and 2, respectively, but with the roles of the [CBPQT-A]⁴⁺/²⁽⁺⁾ and [CBPQT-B]⁴⁺/²⁽⁺⁾ rings reversed. The use of the term 'clockwise' means that each CBPQT⁴⁺ ring visits (Extended Data Fig. 5) the substituents on the loop in the order T/PY⁺/IPP, rather than the other way around.

The [3]catenane was synthesized (Supplementary Scheme 6) by using radical templation[41] leading, in the first instance, to the formation of an intermediate pseudo[3]rotaxane before the final loop closure. The oxidized state **[3]CMM¹³⁺**, which was isolated as its PF₆⁻ salt, was fully characterized (Fig. 2a) by ¹H nuclear magnetic resonance (NMR) spectroscopy. The positions of each CBPQT⁴⁺ ring on the loop were identified by changes in the chemical shifts of proton resonances on the BPM unit (H-13 and H-14) and on the triazole ring (H-28) as a result of the strong shielding by the encircling CBPQT⁴⁺ rings. The reduced state **[3]CMM⁷⁺⁶** of the [3]catenane, which was produced on the addition of 6.0 molar equivalents of cobaltocene[26] (Cp₂Co), or an excess of Zn dust[25], into an MeCN solution of **[3]CMM**·13PF₆ was accompanied by an instantaneous change from colourless to dark purple. The formation of **[3]CMM⁷⁺⁶** can also be followed (Supplementary Fig. 48) by visible/near infrared (Vis/NIR) spectroscopy with a broad absorption band centred on 1,122 nm, which is characteristic[38] of trisradical interactions. The solid-state structure of **[3]CMM⁷⁺⁶** was characterized by X-ray crystallography (Fig. 2b and Supplementary Information Section 7), which confirms the presence of the radical-pairing interactions (Supplementary Fig. 44) between the reduced CBPQT²⁽⁺⁾ rings and the V⁺ units. On addition of NOPF₆, reoxidation of **[3]CMM⁷⁺⁶**, which occurs within seconds, is accompanied by the MeCN solution reverting back to being colourless. The reversible switching between the oxidized and reduced states of the [3]catenane was also investigated (Extended Data Fig. 6 and Supplementary Information Section 8) by cyclic voltammetry, which was carried out on **[3]CMM**·13PF₆ in an MeCN solution. These investigations demonstrated that these two stable states of the [3]catenane can be interconverted using either chemical or electrochemical stimuli.

With the reversible redox switching of the [3]catenane demonstrated, we turned our attention to using electricity to power the molecular motor by controlled potential electrolysis[11,26,41] (CPE). The electrically driven operation (Supplementary Information Section 10) of the molecular motor was accomplished in an electrochemical cell (Extended Data Fig. 7 and Supplementary Video 1) by alternating between two constant potentials (−0.5 V for reduction and +0.7 V for oxidation). The Vis/NIR spectrum of the reduced [3]catenane (30 μM) in MeCN was found to be in excellent agreement (Supplementary Fig. 49) with that obtained from the chemically driven operation with the same concentration (30 μM), confirming the validity of the CPE protocol. We repeated this protocol on the [3]catenane. The whole process was monitored (Fig. 2c) by Vis/NIR spectroscopy. The change in absorbance at 1,122 nm demonstrated (Fig. 2d) that the continuous operation of the motor can be repeated at least five times without any substantial loss in reversibility. The ¹H NMR spectrum (Supplementary Fig. 51) of the final oxidized motor molecule demonstrated recovery of more than 95% of the [3]catenane with negligible degradation, an observation that is commensurate with robust and reliable operation.

By simply accepting the present design of the [3]catenane, it is not possible to investigate the unidirectionality of the motor by determining the positions of the components after one 180° circumrotation on account of the fact that the two CBPQT⁴⁺ rings are constitutionally identical. To address this conundrum, [D₁₆]-**CBPQT⁴⁺** (Supplementary Scheme 7), in which the *p*-xylylene units are deuterium-labelled, was introduced (Extended Data Fig. 8 and Supplementary Scheme 9) during the synthesis of a deuterium-labelled [3]catenane [Dₙ]-**[3]CMM¹³⁺**. As the CBPQT²⁽⁺⁾ ring does not pass over the IPP unit under reducing conditions, there is only one gate (PY⁺ unit) for the CBPQT²⁽⁺⁾ rings threading to form the intermediate pseudo[3]rotaxane. Therefore, the distribution of undeuterated CBPQT⁴⁺ rings on the T and BPM units can be influenced by sequentially adding [D₁₆]-**CBPQT²⁽⁺⁾** and **CBPQT²⁽⁺⁾**. The [3]catenane [Dₙ]-**[3]CMM¹³⁺** was obtained as a mixture of four isotopologues, including [D₀]-**[3]CMM¹³⁺** and [D₃₂]-**[3]CMM¹³⁺**, in addition to two co-constitutional isomers of [D₁₆]-**[3]CMM¹³⁺**. Their presence was confirmed by high-resolution electrospray ionization mass spectrometry (Supplementary Fig. 55) and ¹H NMR spectroscopy (Supplementary Fig. 56). According to the integration of probe resonances for the H-phen′ and H-phen″ in the ¹H NMR spectra, the ratio of

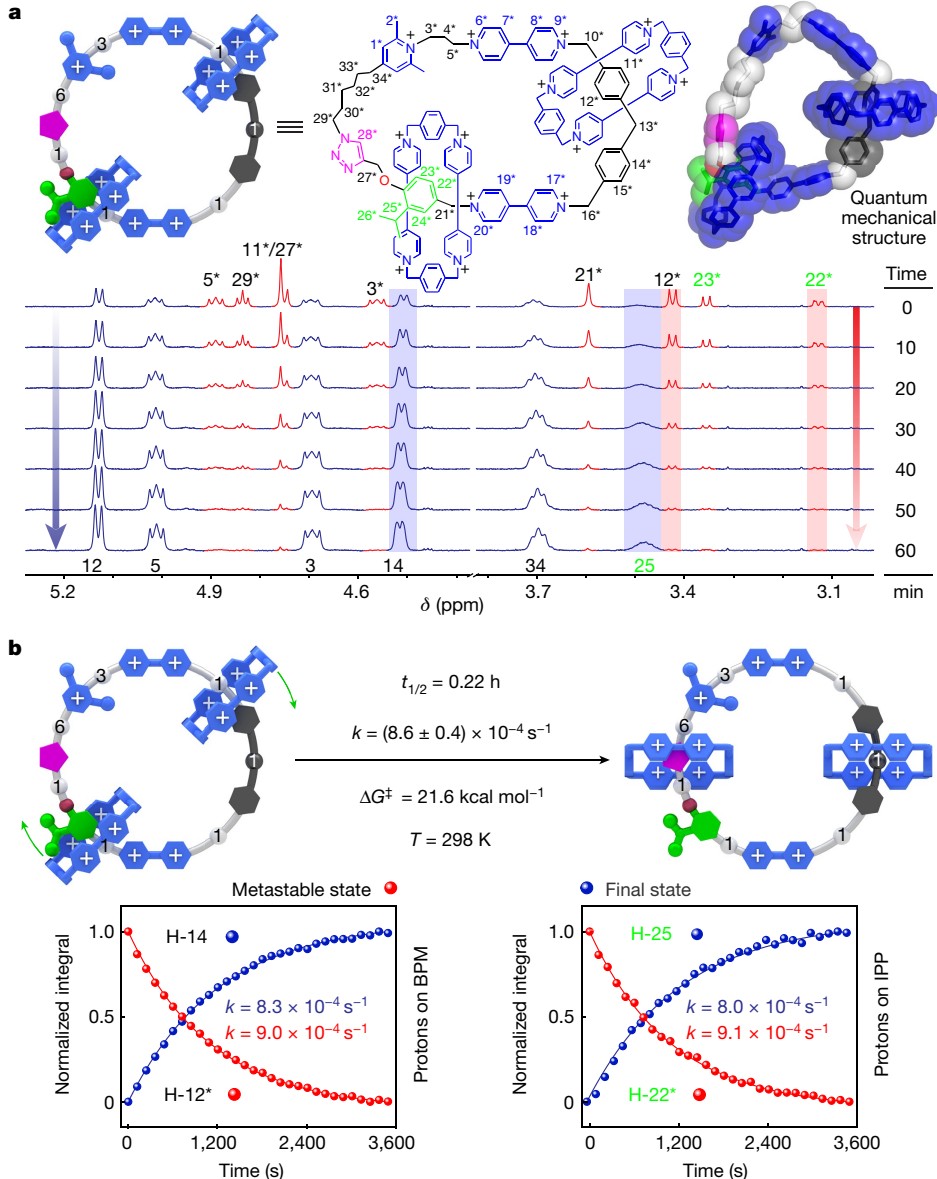

**Fig. 4 | Metastable state in the redox cycle. a**, Top, graphical representation and structural formula of the metastable state with an optimized quantum mechanical model structure (M06-2X/6-31G* basis set, in tubular with superimposed space-filling representation). Bottom, partial $^1$H NMR (600 MHz, CD$_3$CN, 298 K) spectra of **[3]CMM**·13PF$_6$ measured over time (0–60 min) immediately after a cycle of reduction (Cp$_2$Co) and reoxidation (NOPF$_6$), with proton assignments labelled at the top and bottom of the spectra. The proton resonances attributable to the metastable state are labelled with an asterisk. **b**, Top, thermal relaxation associated with the co-conformational rearrangement from the metastable state to the reoxidized state. The activation energy barrier $\Delta G^\ddagger$ of 21.6 kcal mol$^{-1}$ was determined using the Eyring equation ($k = \frac{k_{\mathrm{B}}T}{h}\mathrm{e}^{\frac{-\Delta G^\ddagger}{RT}}$), in which $k$ is the reaction rate constant, $T$ is the absolute temperature, $R$ is the gas constant, $k_{\mathrm{B}}$ is the Boltzmann constant and $h$ is the Planck constant. Bottom, plot of the changes in the normalized integral of protons on the BPM (H-12* and H-14) and IPP (H-22* and H-25) units with time at 298 K during the transformation from the metastable to the reoxidized state, as well as the fitting curves of these data according to the first-order kinetic model. ppm, parts per million.

the undeuterated CBPQT$^{4+}$ rings on the T and BPM units was found to be biased (Supplementary Figs. 56–59), allowing the system as a whole to be monitored. The deuterated motor [D$_n$]-**[3]CMM**$^{13+}$ was subjected to CPE operation to test for the unidirectionality of circumrotation driven by electricity. The unidirectional circumrotation of the rings causes (Fig. 3) co-constitutional exchanges after one redox cycle to be quantified (Extended Data Fig. 9) on the basis of different ratios of integrated values of proton resonances (H-phen' and H-phen") in the $^1$H NMR spectrum. From the $^1$H NMR integrations (Fig. 3), the directionality of the molecular motor was calculated (Extended Data Fig. 10 and Supplementary Information Section 12) to be 85%, that is, 85% of the molecular motors complete a 180° unidirectional (clockwise)

rotation in one redox cycle. Several chemically driven operations of the molecular motor were also performed (Supplementary Figs. 58–61), each providing comparable unidirectionalities of about 85% after one redox cycle.

A metastable state is observed (Fig. 4a, Supplementary Section 13 and Supplementary Fig. 66) during the oxidation process, demonstrating the unidirectional movement of both rings around the loop in the [3]catenane. Comparison (Supplementary Fig. 65) between the $^1$H NMR spectrum of **[3]CMM**·13PF$_6$ and that of the metastable state shows the locations on the loop of two rings. Large upfield shifts are observed (Supplementary Fig. 65) for the resonances associated with the protons belonging to the BPM (H-12*) and the IPP (H-22*) units, whereas

resonances for protons H-14* and H-15* are shifted downfield, indicating (Fig. 4a) that one of the two CBPQT$^{4+}$ rings moves away from the V$^{2+}$ unit and is located asymmetrically on the BPM unit on account of net Coulombic repulsion, whereas the other ring is poised to mount the IPP steric barrier. The positions of the two rings on the loop indicate that the circumrotation towards the metastable state is in the clockwise direction. The subsequent relaxation by clockwise rotation (Extended Data Fig. 10) to the stable oxidized state is favoured. By contrast, the counterclockwise motion going from the metastable state III to the stable oxidized state I would require both of the CBPQT$^{4+}$ rings to pass over the V$^{2+}$ units, which is kinetically much less likely. The appearance of the metastable state and subsequent thermally activated relaxation to the stable oxidized [3]catenane was monitored (Fig. 4a and Supplementary Fig. 76) by $^1$H NMR spectroscopy at 298 K in CD$_3$CN. Kinetic analysis (Fig. 4b and Supplementary Figs. 77 and 78) shows that this co-conformational rearrangement follows first-order kinetics at 298 K with an average rate constant $k$ of $(8.6 \pm 0.4) \times 10^{-4}$ s$^{-1}$, corresponding to an energy of activation ($\Delta G^\ddagger$) of 21.6 kcal mol$^{-1}$. These values are in good agreement with the calculated (Extended Data Fig. 4) value ($\Delta E^\ddagger = 20.1$ kcal mol$^{-1}$). The nearly identical rate constants (Supplementary Figs. 77 and 78) associated with relaxation of the metastable state of each ring suggests that, on oxidation, the two rings undergo unidirectional circumrotation (Supplementary Video 2) on account of the electrostatic interactions arising from the nanoconfinement[46] present in the mechanically interlocked [3]catenane. This mechanism is different from previously reported[4,5] [3]catenane-based molecular motors in that our electric molecular motor is driven by a single stimulus, namely the continuous oscillation of redox potential, that is, an applied direct current voltage.

We often think hierarchically about components of macroscopic machines, in which one component serves to drive another. This kind of hierarchy is impossible in molecular machines, in which movements of components obey the principle of microscopic reversibility. By contrast, the emergent interactions between the two rings in the electric molecular motor that allow for directional motion are reciprocal, but in conjunction with the kinetic asymmetry introduced into the structure of the molecule, they provide a mechanism by which time-symmetric switching between reducing and oxidizing conditions can drive unidirectional rotation. It may well turn out that these noncovalent interactions under nanoconfinement play the role of what is described as 'gearing' in macroscopic machines.

This electrically driven molecular motor exhibits several attractive features that are expected to play an important role in the subsequent development of artificial molecular machines. The [3]catenane can be synthesized in only four steps based on the previously described protocols[40,41]. Moreover, no covalent bonds need to be broken during the unidirectional circumrotation of the two rings. The timescale for the completion of two redox cycles involving the unidirectional circumrotation of the two small rings through 360° with respect to the loop is only a few minutes. Furthermore, the co-constitution of the [3]catenane is compatible with attachment to the surface of an electrode by chemical modification of one of the two small rings, allowing spatially directed rotation with respect to a fixed frame of reference, thereby transducing electrical into mechanical energy at a surface.

The energy ratchet mechanism, on which this electric molecular motor is based, is powered by redox control of the two-dimensional potential energy landscape of the [3]catenane. The energy landscape is sculpted by the design of the constitution of the loop such that externally applied oscillation of the redox potential provides the energy required to harness Brownian motion of the molecular system to achieve unidirectionality. Our design represents a chemical rather than mechanical engineering approach to the development of electric molecular motors[47].

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

## Methods

### Quantum mechanical calculations

To explain the working mechanism of the electric molecular motor, quantum mechanical calculations were performed at the level of density functional theory to study the PESs of the [2]catenanes **[2]C**$^{9+/5+4\bullet}$ and the [3]catenanes **[3]CMM**$^{13+/7+6\bullet}$. In these calculations, the geometries were optimized in the Poisson–Boltzmann solvation model[48] at the level of the M06-2X/6-31G* basis set (ref. [49]) with Jaguar v.10.6 (ref. [50]). Because of the complexity of the systems, all counterions were replaced by an implicit continuous dielectric solvent. To compensate for the effect of counterions, parameters resulting in a stronger solvation effect were necessary. The new solvation parameters were obtained (Supplementary Fig. 36) by fitting the experimentally measured barrier height of the CBPQT$^{4+}$ dethreading from a model pseudorotaxane. The solvation parameters $\varepsilon = 75$ and $R_0 = 1.4$ Å were chosen for the calculations performed on **[2]C**$^{9+/5+4\bullet}$ and **[3]CMM**$^{13+/7+6\bullet}$.

### PESs of [2]catenane

The PESs of **[2]C**$^{9+/5+4\bullet}$ were studied (Extended Data Figs. 1 and 2) by scanning the $z$ coordinate from the centre of CBPQT$^{4+/2(+\bullet)}$, which is defined as the average position of four methylene carbon atoms of CBPQT$^{4+/2(+\bullet)}$, passing over the atoms−labelled as positions 0 to 50−on the loop. The PESs are periodic because of the cyclic nature of the loop.

For the oxidized state **[2]C**$^{9+}$, the PES (Extended Data Fig. 1 and Supplementary Table 1) reaches a maximum (position 31) between the positively charged V$^{2+}$ and PY$^{+}$ units, indicating strong electrostatic repulsion between the CBPQT$^{4+}$ ring and the V$^{2+}$/PY$^{+}$ units on the loop. The other barrier (position 2) is provided by the IPP unit because of the bulky isopropyl group. Whereas the PES shows a minimum (position 0) positioned around the T unit, the other energy well (position 18) is close to the centre of the BPM unit because of donor−acceptor and van der Waals interactions between the CBPQT$^{4+}$ ring and the BPM unit. The energy barriers for the CBPQT$^{4+}$ ring encircling the BPM unit passing over the bulky IPP and the positively charged PY$^{+}$ units are 25.5 and 46.7 kcal mol$^{-1}$, respectively.

For the reduced state **[2]C**$^{5+4\bullet}$, the PES (Extended Data Fig. 2a) reaches a maximum (42.2 kcal mol$^{-1}$) when the CBPQT$^{2(+\bullet)}$ ring passes (position 2) over the bulky IPP unit. All three wells (Extended Data Fig. 2a) result from favourable radical-pairing interactions between the CBPQT$^{2(+\bullet)}$ ring and the V$^{+\bullet}$ units: the first (position 10) and the third (position 27) wells correspond to the CBPQT$^{2(+\bullet)}$ ring encircling the two V$^{+\bullet}$ units (Extended Data Fig. 2d), respectively, whereas the second well (position 21) has a compacted conformation (Extended Data Fig. 2d) because of the radical-pairing interactions between the V$^{+\bullet}$ units and the CBPQT$^{2(+\bullet)}$ ring, which are tilted with respect to each other.

### PESs of [3]catenane

To describe the movement of the two CBPQT$^{4+/2(+\bullet)}$ rings in **[3]CMM**$^{13+/7+6\bullet}$ around the loop, a two-dimensional map was constructed (Extended Data Fig. 5), in which the $x$ and $y$ axes represent the positions of the CBPQT$^{4+/2(+\bullet)}$ rings on the loop. Note that the map is periodic in both dimensions. To simplify the calculations, one of the two CBPQT$^{4+/2(+\bullet)}$ rings (ring A or ring B) was moved to its next position, and then the second ring was allowed to relax to its local minimum. The PESs of **[3]CMM**$^{13+/7+6\bullet}$ on the two-dimensional map were calculated by scanning the $z$ coordinate from the centre of one CBPQT$^{4+/2(+\bullet)}$ ring passing over the labelled atoms on the loop while letting every other degree of freedom, including the position of the other CBPQT$^{4+/2(+\bullet)}$ ring, relax to the local minimum. Eight hypothetical paths were calculated (Supplementary Tables 3–10) for redox switching in the [3]catenanes **[3]CMM**$^{13+/7+6\bullet}$, which are identified in Extended Data Figs. 3b and 4b. In the calculations, it is assumed that all of the reductions and oxidations occur rapidly relative to the ring motion.

In the case of the reduction process (Extended Data Fig. 3), although all four paths experience a decrease in energy at the beginning, path R3 requires passage over the lowest barrier. In path R3, the CBPQT$^{2(+\bullet)}$ ring B was moved first of all to the bottom V$^{+\bullet}$ unit, followed by moving the CBPQT$^{2(+\bullet)}$ ring A to pass (Extended Data Fig. 3e) over the PY$^{+}$ unit to reach the end point II. In comparison, both paths R2 and R4 require the CBPQT$^{2(+\bullet)}$ ring A to pass (Extended Data Fig. 3f) over the bulky IPP unit from point I to II'. As a result, the path-determining energy difference along path R4 is 4.7 kcal mol$^{-1}$ higher than that along path R3. These results indicate that, under reducing conditions, the CBPQT$^{2(+\bullet)}$ rings (1 and 2) strongly prefer to move from point I to II.

The X-ray single-crystal structure (Fig. 2b) of the reduced [3]catenane **[3]CMM**$^{7+6\bullet}$ clearly shows that two CBPQT$^{2(+\bullet)}$ rings encircle the two V$^{+\bullet}$ units in the loop. To study the unidirectional movement of the CBPQT$^{4+}$ rings under oxidizing condition, point II was set to be the starting point. The single-crystal structure was also used as the initial structure for optimizing the geometry of the [3]catenane **[3]CMM**$^{13+}$. On oxidation, the two CBPQT$^{4+}$ rings move away from the V$^{2+}$ units. The question is, which direction is the more favourable one?

Four paths that all start at point II in two different directions were examined (Extended Data Fig. 4). Among the four paths, path O1 has the lowest energy barrier for the CBPQT$^{4+}$ rings moving towards the end point I', in which two CBPQT$^{4+}$ rings encircle the T and BPM units. In path O1, the CBPQT$^{4+}$ ring B was first moved to the BPM unit, followed by moving the CBPQT$^{4+}$ ring A to the T unit, so as to lower the overall energy. By comparison, in path O2, the CBPQT$^{4+}$ ring A was first moved to pass over the PY$^{+}$ unit to its final location at the T unit, followed by moving the CBPQT$^{4+}$ ring A to the BPM unit. The path-determining energy difference along path O2 is 8.1 kcal mol$^{-1}$ higher than that along path O1. Therefore, it is far more favourable for the two CBPQT$^{4+}$ rings to move from point II to point I'.

In conclusion, the quantum mechanical calculations predict unidirectional movement from point I to point II to point I' during the redox cycle of the [3]catenanes **[3]CMM**$^{13+/7+6\bullet}$, which is consistent with the experimental result.

### Cyclic voltammetry

To gain a better understanding of the electron-transfer processes during the redox cycle undergone by the [3]catenane **[3]CMM**$^{13+}$, variable scan-rate cyclic voltammetry experiments (Extended Data Fig. 6) were performed.

The cyclic voltammetry profile shows (Extended Data Fig. 6) three reduction peaks with potentials at −0.08 V, −0.15 V and −0.25 V at a low scan rate (0.02 V s$^{-1}$), corresponding to reduction associated with radical formation, starting from **[3]CMM**$^{13+}$ and leading to the production of **[3]CMM**$^{7+6\bullet}$. The first two reduction peaks (−0.08 V and −0.15 V) account for the stepwise formation of viologen radical pairs as a consequence of the different chemical environments experienced by the two CBPQT$^{4+}$ rings in the [3]catenane **[3]CMM**$^{13+}$. The first reduction peak at −0.08 V can be assigned to the reduction of one of the V$^{2+}$ units in the loop and one of the two V$^{2+}$ units in the CBPQT$^{4+}$ ring encircling the BPM unit, resulting in a decrease in Coulombic repulsion, while establishing stabilizing radical-pairing interactions between the mechanically interlocked components. The following reduction peak at −0.15 V can be attributed to the reduction of the other V$^{2+}$ unit in the loop and one of the two V$^{2+}$ units in the other CBPQT$^{4+}$ ring encircling the T unit. The third reduction peak observed at −0.25 V, corresponding to two simultaneous one-electron reductions, accounts for the further reduction of both CBPQT$^{2+(+\bullet)}$ monoradical trication rings to their diradical dicationic states CBPQT$^{2(+\bullet)}$. The oxidation of the radical state **[3]CMM**$^{7+6\bullet}$ back to **[3]CMM**$^{13+}$ occurs in two steps at −0.16 V and +0.08 V. The first oxidation peak at −0.16 V can be assigned to two one-electron oxidations of the two unpaired V$^{+\bullet}$ units in the two CBPQT$^{(2+\bullet)}$ rings (CBPQT$^{2(+\bullet)} \to$ CBPQT$^{2+(+\bullet)}$), resulting in much weaker binding interactions and increased Coulombic repulsion between

the components ($V^{+\bullet}$ and $CBPQT^{2+(+\bullet)}$), that is, **[3]CMM$^{7+6\bullet}$** is oxidized to **[3]CMM$^{5+4(+\bullet)}$**. It is followed by four simultaneous one-electron oxidations, namely two $V^{+\bullet}$ units in the two $CBPQT^{2+(+\bullet)}$ rings and two $V^{+\bullet}$ units in the loop are oxidized at the same oxidation potential (+0.08 V), resulting in the formation of the fully oxidized state **[3]CMM$^{13+}$**. These observations are consistent with previously published results[51].

The reduction of the radical state **[3]CMM$^{7+6\bullet}$** to its neutral (viologen) form **[3]CMM$^+$** involves (Supplementary Fig. 46) three sequential two-electron reversible processes. The first and less negative one (−0.77 V, peak potential) can be assigned to the reduction of the two unpaired $V^{+\bullet}$ units in the two $CBPQT^{2(+\bullet)}$ rings, leading to the formation of **[3]CMM$^{+4(+\bullet)}$**, which is stabilized[52] by both radical-pairing and donor−acceptor interactions. The following two-electron process accounts for the reduction of the other two $V^{+\bullet}$ units in the two $CBPQT^{(+\bullet)}$ rings. Finally, the reduction of the remaining two $V^{+\bullet}$ units in the loop occurs at −1.01 V, reflecting the presence of mechanical bonding and the nanoconfined geometry of the [3]catenane.

As the scan rate is increased to 2.0 V s$^{-1}$, only one reduction peak for the radical state **[3]CMM$^{7+6\bullet}$** is observed, indicating that the electron-transfer process is much faster than ring movement at this fast scan rate. This observation suggests that, under the experimental conditions used during the operation of the [3]catenane motor, the reduction to the radical state **[3]CMM$^{7+6\bullet}$** and the reoxidation to the fully oxidized state **[3]CMM$^{13+}$** is completed fully and very rapidly.

### Electrically driven operation of [3]CMM

Electrically driven operation of the [3]catenane molecular motor **[3]CMM** was conducted in the $N_2$-filled glovebox. An MeCN (38 ml, 0.1 M TBAPF$_6$) solution of **[3]CMM·13PF$_6$** (30 μM) was added to a BASi bulk electrolysis cell, which was equipped (Extended Data Fig. 7) with a reticular vitreous carbon working electrode, a coiled platinum-wire auxiliary electrode within a fritted glass chamber and a Ag/AgCl reference electrode, and connected to a Gamry multipurpose instrument (Reference 600) interfaced to a PC. The auxiliary electrode chamber was filled with an excess of Cu(MeCN)$_4$PF$_6$ dissolved in MeCN (1 ml, 0.1 M TBAPF$_6$). The auxiliary electrode constituted a platinum wire wrapped with a copper wire (diameter 0.25 mm, 99.999% trace metals basis from Sigma-Aldrich). The experimental parameters were controlled using the software of Gamry Framework v.6.30 operating in the chronocoulometry mode. A less negative reduction potential (−0.5 V) and a less positive oxidation potential (+0.7 V) were used to limit the degradation of the [3]catenane. The whole apparatus was subjected to five redox cycles with alternate constant potentials of −0.5 V (reduction potential versus Ag/AgCl) and +0.7 V (oxidation potential versus Ag/AgCl) for 10 and 15 min, respectively.

### Data availability

The data that support the findings of this study are available within the paper and its Supplementary Information files. Crystallographic data for the [3]catenane in its reduced state **[3]CMM$^{7+6\bullet}$** can be obtained free of charge from www.ccdc.cam.ac.uk under CCDC deposition number 2168726.

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

**Acknowledgements** We thank Northwestern University (NU) for its support of this research, the Integrated Molecular Structure Education and Research Center (IMSERC) at NU for providing access to equipment for relevant experiments, S. Shafaie for assistance with high-resolution mass spectrometry measurements, Y. Zhang and Y. Wu for help with NMR spectroscopic measurements and C. Cheng for useful discussions. The computational investigations at California Institute of Technology were supported by National Science Foundation grant no. CBET-2005250 (W.-G.L. and W.A.G.).

**Author contributions** J.F.S. directed the project. L.Z. and C.P. conceived the idea for the project. L.Z. designed, synthesized and characterized the compounds. W.-G.L. and W.A.G. performed density functional theory calculations. R.D.A. contributed to theoretical analyses carried out on the mechanism of operation of the electric molecular motor. Y.Q. provided some of the precursors and performed CPE experiments. H.C. contributed to the graphical design used in the figures. D.S. conducted the video recording and contributed to the animation. B.S. carried out electrospray ionization mass spectrometry and travelling-wave ion mobility mass spectrometry measurements and analyses. C.L.S. collected the single-crystal X-ray diffraction data and solved the solid-state structure. K.C., H.W., Y.J., Y.F., J.S.W.S., J.T., Y.T., X.-Y.C., Q.-H.G. and D.P. commented on the data. All the authors participated in evaluating the results. L.Z., R.D.A. and J.F.S. produced numerous drafts of the manuscript and supplementary materials, with input from all authors.

**Competing interests** The authors declare no competing interests.

**Additional information**
**Correspondence and requests for materials** should be addressed to Long Zhang, R. Dean Astumian, William A. Goddard or J. Fraser Stoddart.

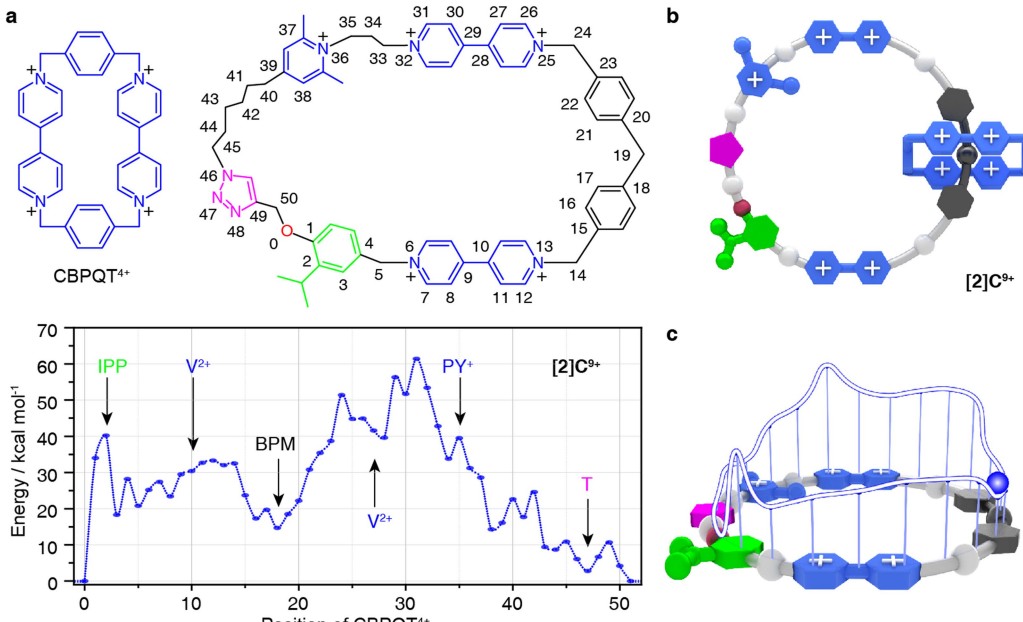

**Extended Data Fig. 1 | Calculated PES of the oxidized [2]catenane [2]C⁹⁺.**
**a**, The potential energy for the CBPQT⁴⁺ ring traversing the loop in the oxidized state. The numbered atoms on the loop are used to define the position of the CBPQT⁴⁺ ring. **b**, Graphical representation of the oxidized state of the [2]catenane **[2]C⁹⁺**. **c**, Graphical representation of the calculated PES of the CBPQT⁴⁺ ring moving around the loop, shown in a rollercoaster manner for the fully oxidized **[2]C⁹⁺**.

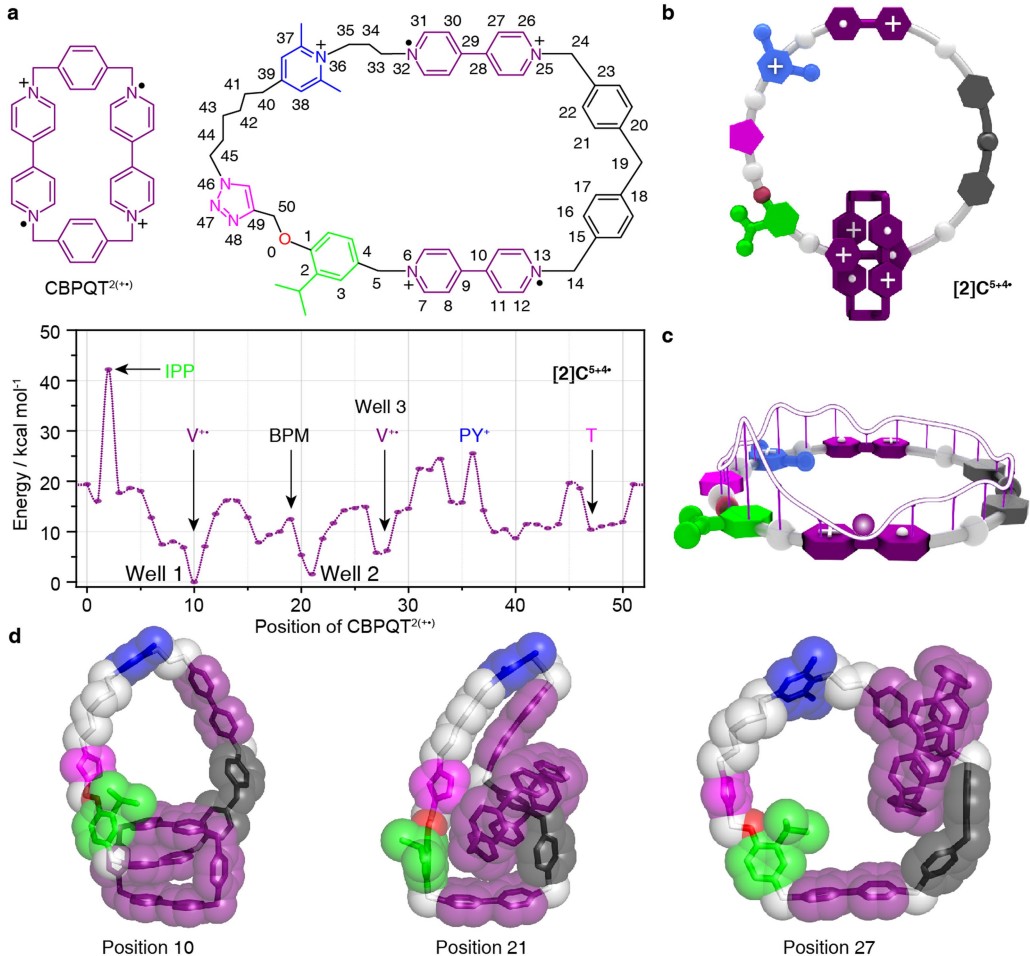

**Extended Data Fig. 2 | Calculated PES of the reduced [2]catenane [2]C$^{5+4•}$.**
**a**, The potential energy for the CBPQT$^{2(+•)}$ ring as a function of its position on the loop in the reduced state of the [2]catenane **[2]C$^{5+4•}$**. The numbered atoms on the loop are used to define the position of the CBPQT$^{2(+•)}$ ring. **b**, Graphical representation of the reduced state of [2]catenane **[2]C$^{5+4•}$** in its lowest-energy co-conformation in which the CBPQT$^{2(+•)}$ ring encircles the V$^{+•}$ unit, position 10. **c**, Graphical representation of calculated potential energy of the CBPQT$^{2(+•)}$ ring as it moves around the loop, shown in a rollercoaster manner for the radical state **[2]C$^{5+4•}$**. **d**, Quantum mechanical minimized structures (M06-2X/6-31G* basis set) for the CBPQT$^{2(+•)}$ ring located at positions 10, 21 and 27, respectively.

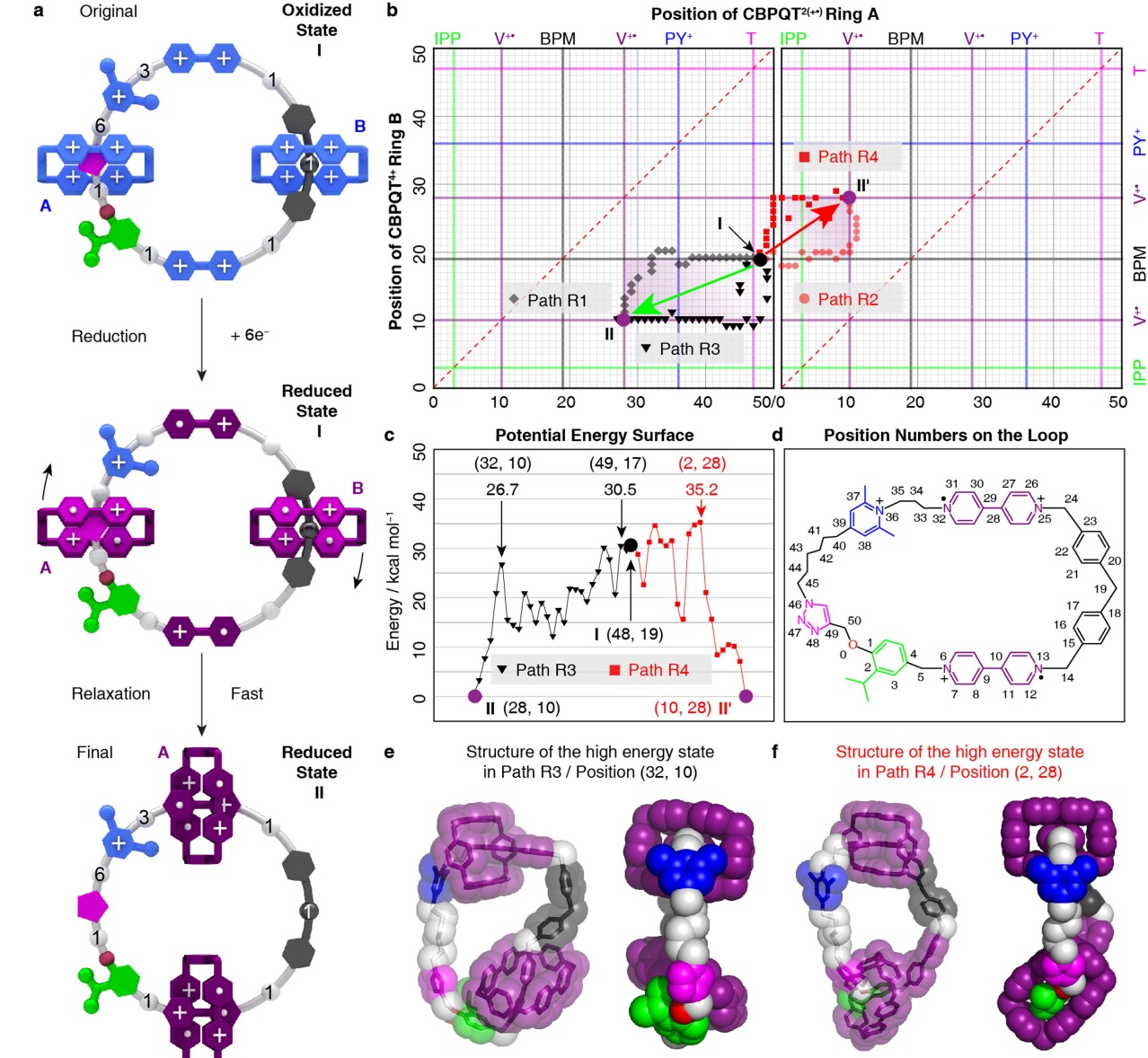

**Extended Data Fig. 3 | Calculated PES of the reduced [3]catenane [3]CMM[7+6].**
**a**, Graphical representations of the redox process experienced by the [3]catenane in going from the oxidized state I (48, 19) to the reduced state II (28, 10). **b**, A two-dimensional position map describing the movement of the two reduced CBPQT[2(+•)] rings (A and B) around the loop. Four hypothetical paths (R1–R4) during the reduction process are illustrated by diamond (grey), circle (light red), triangle (black) and square (red) symbols, respectively. The green arrow indicates the preferred direction of movement and the red arrow indicates the less preferred (nearly precluded) direction of movement. The green, purple,

black, blue and magenta lines represent the positions of the IPP, V[+•], BPM, PY[+] and T units, respectively. The dashed red diagonal lines represent barriers that cannot be crossed physically because doing so would require the two CBPQT[2(+•)] rings to occupy the same space. **c**, The PESs of the two CBPQT[2(+•)] rings moving around the loop in the reduced state, starting from point I (48, 19) and following paths R3 and R4, respectively. **d**, Structural formula of the loop with atoms numbered to define the positions of the CBPQT[2(+•)] rings. **e,f**, Quantum mechanical minimized structures (M06-2X/6-31G* basis set, top-down and side-on views) for the CBPQT[2(+•)] rings at positions (32, 10) and (2, 28), respectively.

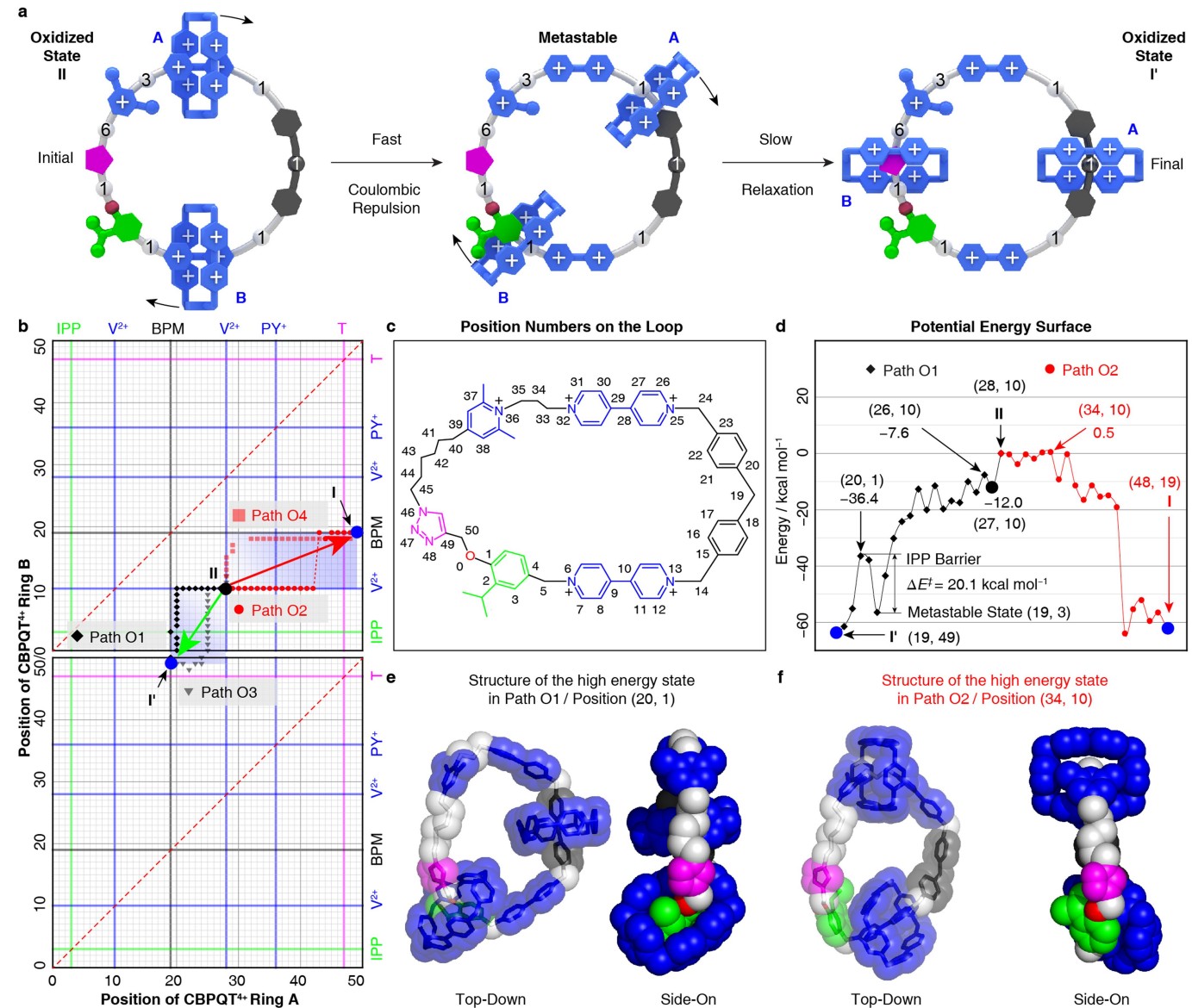

**Extended Data Fig. 4 | Calculated PES of the oxidized [3]catenane [3]CMM¹³⁺.**
**a**, Graphical representations of the process experienced by the [3]catenane in going from the oxidized state II (28, 10) to the oxidized state I' (19, 49). **b**, A two-dimensional position map describing the movement of the two oxidized CBPQT⁴⁺ rings (A and B) around the loop. Four hypothetical paths O1–O4 during the oxidation process are illustrated by diamond (black), circle (red), triangle (grey) and square (light red) symbols, respectively. The green arrow indicates the preferred direction of movement and the red arrow indicates the less preferred (nearly precluded) direction of movement. The green, blue, black, blue and magenta lines represent the positions of the IPP, V²⁺, BPM, PY⁺ and T units, respectively. The dashed red diagonal line represents a barrier that cannot be crossed physically because the two CBPQT⁴⁺ rings would occupy the same space. **c**, Structural formula of the loop with atoms numbered to define the positions of the CBPQT⁴⁺ rings. **d**, The PESs of the two CBPQT⁴⁺ rings moving around the loop in the oxidized state starting from point II (28, 10) and following paths O1 and O2, respectively. The value for the energy barrier $\Delta E^{\ddagger}$ of 20.1 kcal mol⁻¹ was determined from the energy difference between the positions (19, 3) and (20, 1). The position (19, 3) corresponds to the metastable state on path O1. **e**,**f**, Quantum mechanical minimized structures (M06-2X/6-31G* basis set, top-down and side-on views) for the CBPQT⁴⁺ rings at positions (20, 1) and (34, 10), respectively. Quantum mechanical minimized structures of the lowest-energy state (19, 48) and metastable state (21, 3) are presented in Figs. 2a and 4a, respectively.

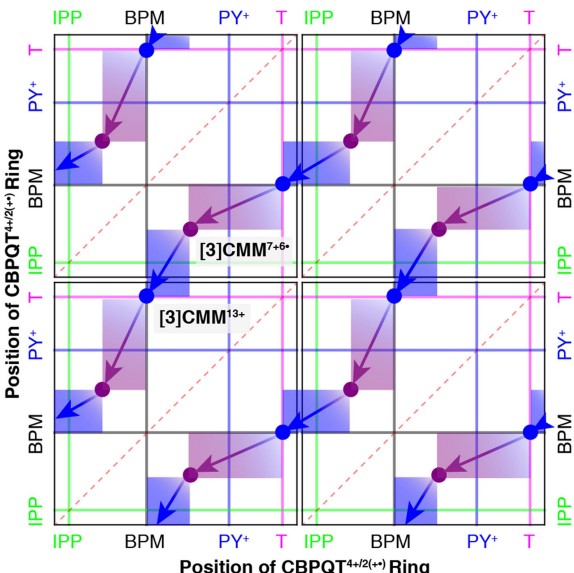

**Extended Data Fig. 5 | A two-dimensional position map describing the movement of two CBPQT$^{4+/2(++)}$ rings around the loop.** The $x$ and $y$ axes represent the positions of the CBPQT$^{4+/2(++)}$ rings on the loop. The green, black, blue and magenta lines represent the positions of the IPP, BPM, PY$^+$ and T units, respectively. The hypothetical paths involving switching of the [3]catenane **[3]CMM** during the redox cycles are illustrated in blue (oxidation) with purple (reduction) arrows, which are periodic on account of the cyclic nature of the loop. The trajectories indicate that, for unidirectional motion, there is a Coulombic barrier (PY$^+$) and a steric barrier (IPP) under reducing and oxidizing conditions, respectively. The dashed red diagonal line represents a barrier that cannot be crossed physically for the simple reason that two CBPQT$^{4+/2(++)}$ rings would end up occupying the same space.

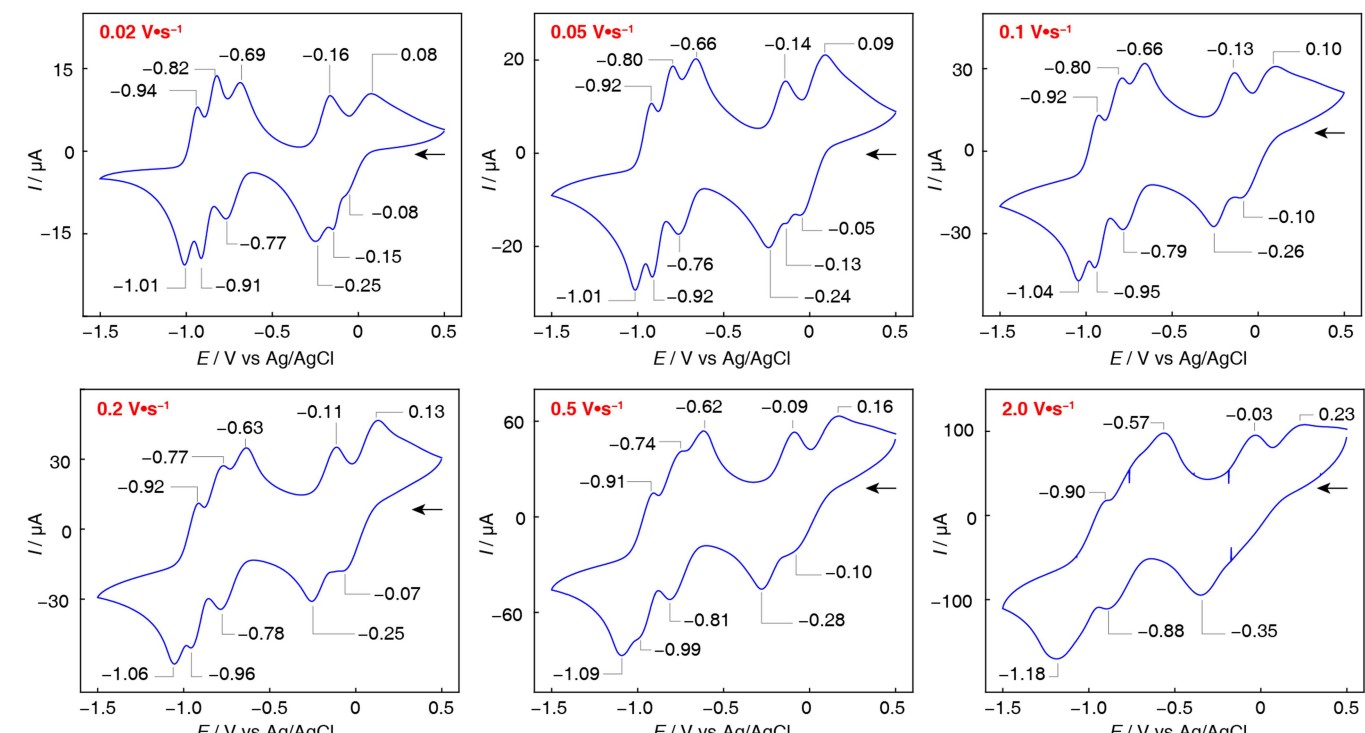

**Extended Data Fig. 6 | Scan-rate variation (0.02–2.0 V s⁻¹) of cyclic voltammograms of [3]CMM•13PF₆ (0.5 mM).** As the scan rate is increased to 2.0 V s⁻¹, only one reduction peak for the radical state **[3]CMM⁷⁺⁶•** is observed, indicating that the electron-transfer process is much faster than ring movement at this fast scan rate. This observation suggests that, under the experimental conditions used during the operation of the [3]catenane motor, the reduction to the radical state **[3]CMM⁷⁺⁶•** and the reoxidation to the fully oxidized state **[3]CMM¹³⁺** is completed fully and very rapidly.

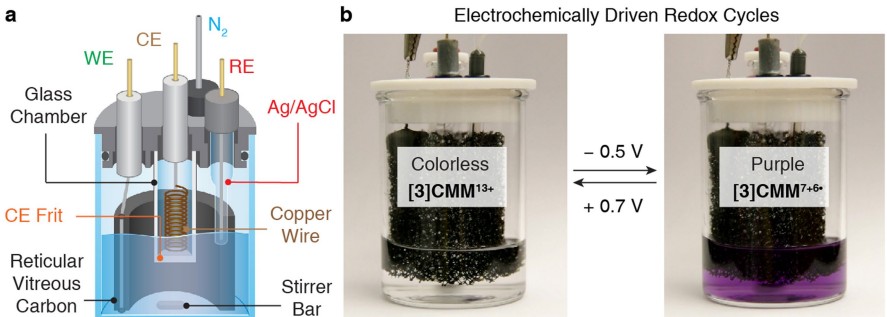

**Extended Data Fig. 7 | Electrically driven operation of [3]CMM. a**, Graphical illustration of the electrochemical cell used in the repeated controlled potential electrolysis experiments. CE, counter electrode; RE, reference electrode; WE, working electrode. **b**, Photographs of the oxidized state (left) **[3]CMM$^{13+}$** and the reduced state (right) **[3]CMM$^{7+6\cdot}$** in the electrochemical cell.

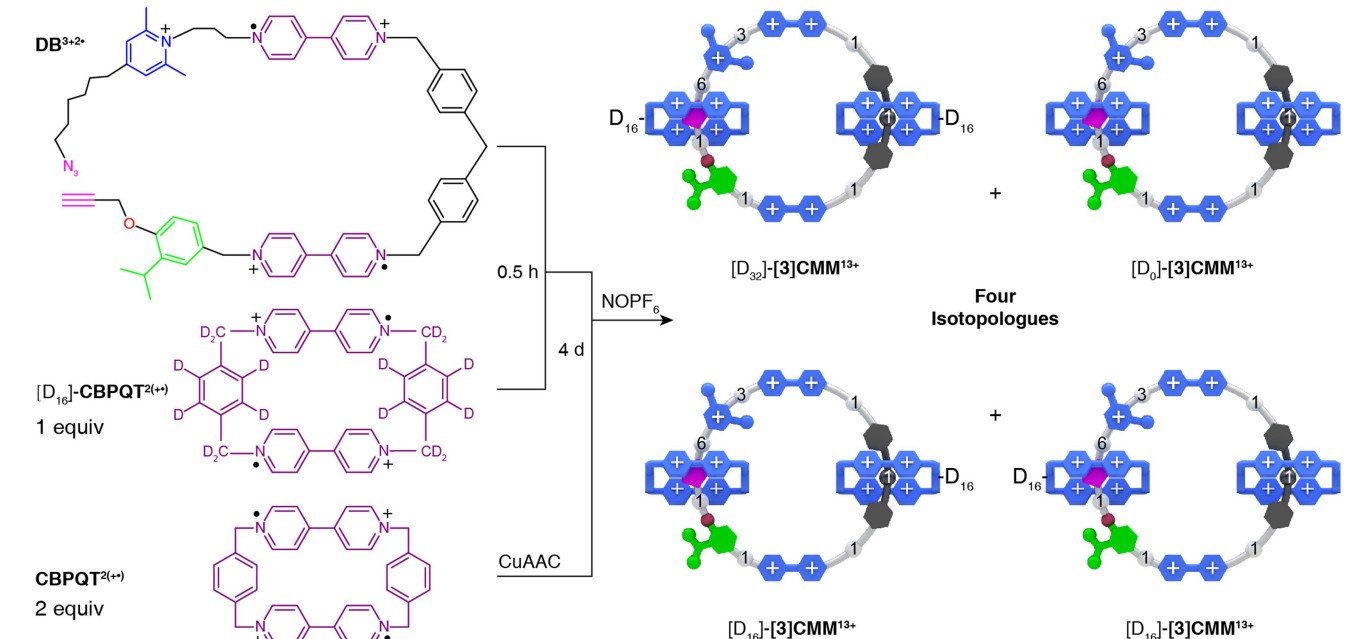

**Extended Data Fig. 8 | Synthesis of the deuterium-labelled [3]catenane [D_n]-[3]CMM.** By sequentially adding [D$_{16}$]-**CBPQT$^{2(+\bullet)}$** and **CBPQT$^{2(+\bullet)}$** during the synthesis of **[D$_n$]-[3]CMM**, it is able to bias the distribution of the undeuterated CBPQT$^{4+}$ rings on the T and BPM units. The [3]catenane [D$_n$]-**[3]CMM$^{13+}$** was obtained as a mixture of four isotopologues, including [D$_0$]-**[3]CMM$^{13+}$**, and [D$_{32}$]-**[3]CMM$^{13+}$**, in addition to two co-constitutional isomers [D$_{16}$]-**[3]CMM$^{13+}$**. CuAAC, copper(I)-catalysed azide–alkyne cycloaddition.

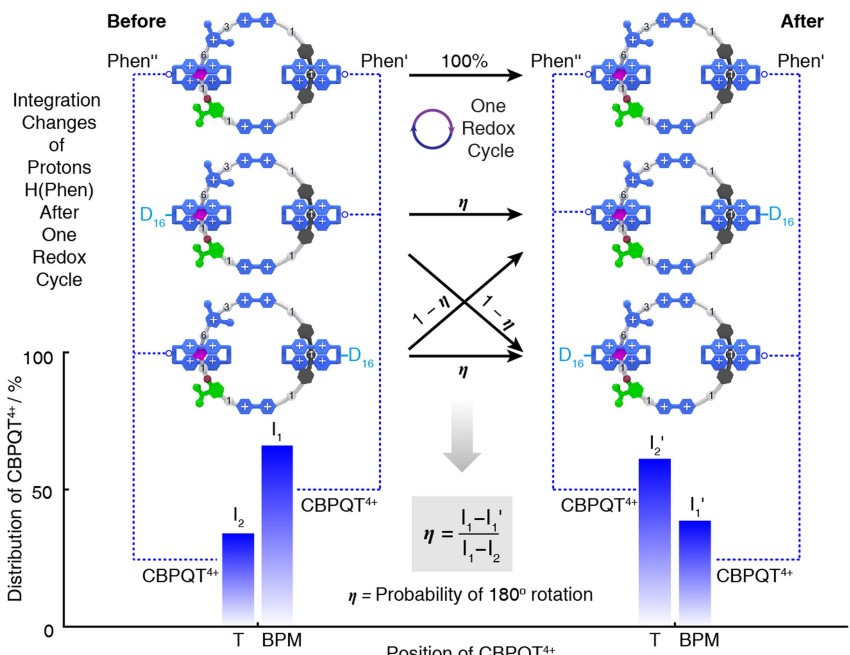

**Extended Data Fig. 9 | Calculation of the probability of 180° rotation for both rings on the loop.** Distribution of CBPQT$^{4+}$ rings on the two positions (T and BPM) of the cyclic track according to the $^1$H NMR spectra (Fig. 3) of [D$_n$]-**[3]CMM**·13PF$_6$ before and after one redox cycle, based on the relevant integration of protons H-Phen″ (I$_1$/I$_1$′) and H-Phen′ (I$_2$/I$_2$′), respectively. $\eta$, the probability of 180° rotation for both rings on the loop.

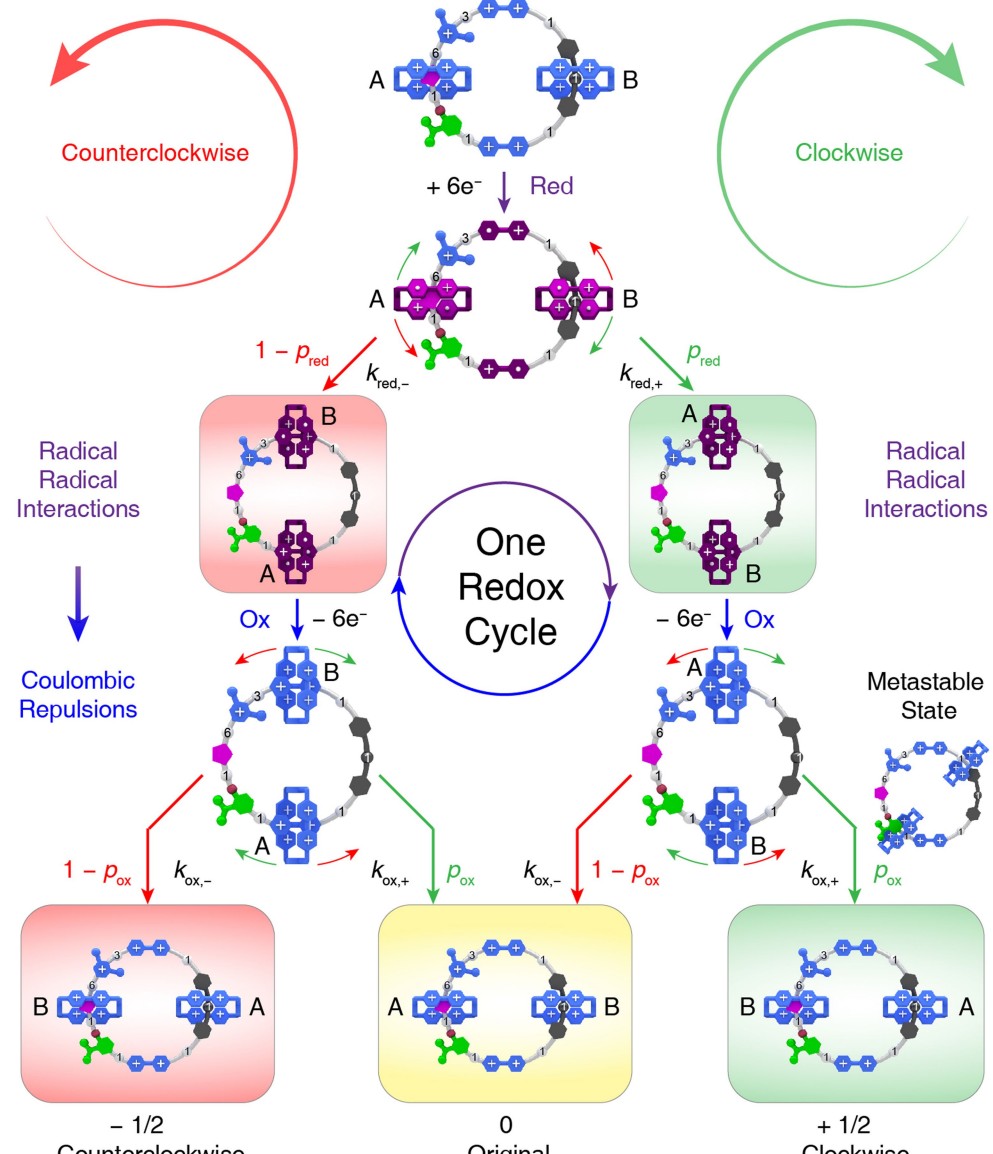

**Extended Data Fig. 10 | Probable direction of the movement of two CBPQT$^{4+/2(+•)}$ rings during the redox operation of [3]CMM$^{13+}$.** Clockwise means that each CBPQT$^{4+/2(+•)}$ ring sees the substituents on the loop in the order T → PY$^+$ → V$^{2+/+•}$ → V$^{2+/+•}$ → IPP → T and counterclockwise means that each ring sees the substituents on the loop in the order T → IPP → V$^{2+/+•}$ → V$^{2+/+•}$ → PY$^+$ → T. The curly arrows represent the direction in which the two rings move around the loop. The '$p_{red}$' and '$p_{ox}$' near the arrows are the probability of clockwise motion of the two rings on the loop in the redox cycle. The $k_{red,+}$, $k_{red,-}$, $k_{ox,-}$ and $k_{ox,+}$ are the corresponding rate constants for the probable steps during the redox cycle, respectively, and the clockwise and counterclockwise steps are also indicated by '+' and '−', respectively. The 180° rotation is represented by '1/2'.