## [Peer Review File · Nature]

Peer

Review File Manuscript Title: An electric

molecular motor.

Reviewer Comments & Author Rebuttals

Reviewer Reports on the Initial Version:

Referees' comments:

Referee #1 (Remarks to the Author):

The authors present a molecular motor based on a [3]catenane architecture. The motor comprises of two cyclobis(paraquat-*p*-phenylene) (CBPQT⁴⁺) rings mechanically interlocked around a track consisting of multiple recognition sites and steric/electronic barriers. The motor employs an energy ratchet design, reminiscent of the strategy employed by Leigh in 2003,¹ where one macrocycle blocks the backwards movement of the other. The system is powered by the Stoddart group's well-established 2013 redox-driven energy ratchet (dubbed a pumping cassette).² The redox ratchet has seen multiple rounds of iteration, having been optimised to its current form in 2015³ and having been demonstrated to work electrochemically in 2018,⁴ as such, the system is robust and well-understood, and thus provides a good basis for the development of this new motor. As with these previous related systems, successive reduction and oxidation of the CBPQT⁴⁺ rings and viologen binding sites allows directional movement of the rings. The macrocycles preferentially pass over a pyridinium cationic barrier towards the viologen^{+•} binding sites with CBPQT^{2(+•)} in the reduced diradical dicationic form, but over a isopropylphenylene steric barrier towards alternative binding sites when reoxidised to CBPQT⁴⁺. As opposed to previous versions, which progressively pump macrocycles out of solution onto a thread,³⁻⁹ or along a linear track,^{2,9,10} this iteration of the ratchet is used to control the rotary motion of the CBPQT⁴⁺ rings around a circular track.

The authors demonstrate the use of controlled potential electrolysis to operate this motor using an electrical current, affording a non-invasive, repeatable and clean (in that no waste products are formed) method for reliable operations. Alternatively, the authors show that the motor can also be successfully operated by a wide array of chemical reductants and oxidants. Finally, the authors investigate the directionality of the motor, using a deuterium labelled CBPQT⁴⁺ macrocycle to establish that the rings swap positions during a redox cycle. 85% of motors complete a 180° clockwise rotation per redox cycle. This conclusion is supported by a series of kinetic models and computational simulations of the potential energy surface.

The structure of the report is logical, the experimental work is well presented and the supplementary information is detailed and justifies their observations. However, I have a few concerns about the discussion that should be clarified:

1. The authors describe their motor as operating by a combination of an energy ratchet, information ratchet. This seems incorrect to me, and indeed the motor appears to operate purely by an energy ratchet mechanism.¹¹ In the operation cycle of the motor, oxidation or reduction breaks the balance of the system, with the macrocycles placed in a thermodynamically unfavoured state. The macrocycles subsequently relax to a local (kinetically defined) equilibrium position via the most kinetically accessible pathway, constituting a power stroke.

By contrast, in an information ratchet,^{11,12} the key component is the kinetic gating of the chemical, photochemical, or electrochemical reactions that switch the ratchet between states, such that the reaction occurs more rapidly when it enables forward movement (as is well explained in the authors' previous work).¹³

It is possible that some element of information ratchet mechanism may be present if the rate of reduction occurs more rapidly with CBPQT⁴⁺ over the triazole

and bis(4- methylene-phenyl)methane stations and/or that the rate of oxidation occurs more rapidly when CBPQT^{2(+•)} binds the viologen⁺ units. This must then result in a kinetic bias for forward movement. No such evidence is currently provided, and I recommend the authors remove this claim unless they can provide such evidence.

2. The mechanism of operation of the motor closely reflects Leigh's 2003 [3]catenane¹ (which operates purely by an energy ratchet mechanism). I consider that the authors should explicitly highlight the connection to this previous work, particularly the use of a second macrocycle to block backwards motion of the first.
3. The authors could also more explicitly highlight their own papers that this new system builds on,²⁻¹⁰ especially papers detailing the origin of the techniques used for this motor.²⁻⁵
4. Feringa has previously demonstrated an electrically-driven molecular motor, which should be cited.¹⁴ The authors might also relate their work to the electrically driven motors of Sykes.¹⁵ In both cases, the authors may wish to highlight the substantial differences between their system and this previous work.
5. The authors could emphasise and expand upon the key novel aspects of their system, such as the advantages of the rapid and clean electrochemical operation and good directional fidelity.
6. The analogy between the reported system and molecular gears feels rather forced to me. I'm not sure that it is either relevant nor valid and it could be removed entirely. The term 'Molecular Gears' is generally used to describe intermeshed components, whereby rotation about an axis of one component induces a rotation about the axis for another component. This system does not utilise interaction.¹⁶
7. The authors state in the conclusion that the motor operates "continuously". As the conditions must be switched back and forth between reductive and oxidative, this is misleading as it implies that only a single set of conditions are needed for operation (as for light driven motors^{17,18} or chemically fuelled information ratchet motors^{19,20}). The authors should clarify that the motor operates by *continuous oscillation* of redox potential.

References

1. Leigh, D. A., Wong, J. K. Y., Dehez, F. & Zerbetto, F. Unidirectional rotation in a mechanically interlocked molecular rotor. *Nature* **424**, 174–179 (2003).
2. Li, H. et al. Relative unidirectional translation in an artificial molecular assembly fueled by light. *J. Am. Chem. Soc.* **135**, 18609–18620 (2013).
3. Cheng, C. et al. An artificial molecular pump. *Nat. Nanotechnol.* **10**, 547–553 (2015).
4. Pezzato, C. et al. Controlling dual molecular pumps electrochemically. *Angew. Chem. Int. Ed.* **57**, 9325–9329 (2018).
5. Qiu, Y. et al. A precise polyrotaxane synthesizer. *Science* **368**, 1247–1253 (2020).
6. Feng, L. et al. Active mechanisorption driven by pumping cassettes. *Science* **374**, 1215–1221 (2021).
7. Li, X. et al. Fluorescence quenching by redox molecular pumping. *J. Am. Chem. Soc.* **144**, 3572–3579 (2022).
8. Cai, K. et al. Molecular-pump-enabled synthesis of a daisy chain polymer. *J. Am. Chem. Soc.* **142**, 10308–10313 (2020).
9. Guo, Q.-H. et al. Artificial Molecular Pump Operating in Response to Electricity and Light. *J. Am. Chem. Soc.* **142**, 14443–14449 (2020).
10. Qui, Y. et al. A molecular dual pump. *J. Am. Chem. Soc.* **141**, 17472–17476 (2019).
11. Kay, E. R., Leigh & D. A., Zerbetto F., Synthetic molecular motors and mechanical machines. *Angew. Chem. Int. Ed.* **46**, 72–191 (2007).
12. Serreli, V., Lee, C.-F., Kay, E. R. & Leigh, D. A. A molecular information ratchet. *Nature* **445**, 523–527 (2007).
13. Astumian, R. D. Kinetic asymmetry allows macromolecular catalysts to drive an

- information ratchet. *Nat. Commun.* **10**, 3837 (2019).
14. Kudernac, T., et al. Electrically driven directional motion of a four-wheeled molecule on a metal surface. *Nature* **479**, 208–211 (2011).
 15. Tierney, H. L. et al. Experimental demonstration of a single-molecule electric motor. *Nat. Nanotechnol.* **6**, 625–629 (2011).
 16. Gisbert, Y., Abid, S. Kammerer, C. & Rapenne, G. Molecular gears: from solution to surfaces. *Chem. Eur. J.* **27**, 12019–12031 (2021).
 17. Pooler, D. R. S., Lubbe, A. S., Crespi, S. & Feringa, B. L. Designing light-driven rotary molecular motors. *Chem. Sci.* **12**, 14964–14986 (2021).
 18. Koumura, N., Zijlstra, R. W. J., van Delden, R. A., Harada, N. & Feringa, B. L. Light-driven monidirectional molecular rotor. *Nature* **401**, 152–155 (1999).
 19. Wilson, M. R., Solà, J., Carlone, A., Goldup, S. M., Lebrasseur, N. & Leigh, D. A. An autonomous chemically fuelled small-molecule motor. *Nature* **534**, 235–240 (2016).
 20. Borsley, S., Kreidt, E., Leigh, D. A. & Roberts, B. M. W. Autonomous fuelled directional rotation about a covalent single bond. *Nature* **604**, 80–85 (2022).

Referee #2 (Remarks to the Author):

424641_0

Stoddart et al reported an electric molecular motor with unidirectional rotary motion. First of all, this reviewer is convincing the authors' conclusions that this molecular motor exhibits the unidirectional movement driven by the redox cycles. The experiments with using [D_n]-[3]CMM•13PF₆ (Fig. 4e) elegantly demonstrates the unidirectional rotary motion (85% probability for clockwise rotation). All experiments were well-designed and performed with high degree of perfection (including reference citations). Thus, the scientific value of this contribution is worth to be published in *Nature*. The authors are advised to address the following suggestions and questions, if the suggested discussions provide the additional scientific value in this significant work.

(1) Regarding to rotation of D-labeled CBPQT⁴⁺ upon more than two redox cycles:

In Supplementary Fig. 45, the authors successfully provided ¹H NMR spectrum of [3]CMM•13PF₆ after five redox cycles, and confirmed the good reversibility of the rotation of CBPQT⁴⁺ rings. Conversely, in the case of the D-labeled [3]CMM•13PF₆, ¹H NMR spectra only after one redox cycles (corresponding to 180 degree rotation) were given in Fig. 4e and Supplementary Figs. 52 and 53. Could the authors provide ¹H NMR spectra of [D_n]-[3]CMM•13PF₆ after more than two redox cycles? I would like to see the positional exchange between Phen' and Phen'' after more than the 360 degree rotation. The ratio between Phen' (deuterated-poor ring) and Phen'' (deuterated-rich ring) would be changed depending on the number of redox cycle. Here, I shall distinguish the two CBPQT⁴⁺ rings as rings A and B,

respectively. In order to aid the discussion, the rotation is started from [ring A]:[ring B] = 100:0 and 0:100 at $V1^{2+}$ and $V2^{2+}$, respectively (Figure R1). In accordance to the 85% probability for clockwise rotation, the ratio between rings A and B can change depending on the number of redox cycle to approach [ring A]:[ring B] = 50:50. Accordingly, the electric molecular motor can record history of the number of rotations in statistical meaning (using huge number of molecules). This could not be achieved by the macroscopic motor (normally used solely) with perfect unidirectional rotation. This is, for me, interesting.

Figure R1.

(2) Regarding to the electron-transfer mechanism (origin of metastable state of

[3]CMM¹³⁺) Most probably, the authors consider the electron transfer process like vertical transition as shown in Figure R2 (mechanism A), where therefore the energetically unstable species with high electrostatic repulsion (oxidized state II in Supplementary Fig. 40a) was assumed. In accordance with the theory (while formally 6e⁻ electron transfer), electron transfer coupled with partial rotation of the two rings would be more straightforward (mechanism B in Figure R2), where the reduced- and oxidized-states have the same reaction coordinate at the closing point (reduced state X and oxidized state X). I understand that this point has no huge impact on the conclusion of unidirectional rotation. However, the electron-transfer mechanism may be important for interpreting the origin of metastable state

[3]CMM¹³⁺ (: when the two rings start to move). Please consider (discuss) the possibility of mechanism B, and tell me the reason which is more convincing mechanism in this case. In addition, does the metastable state [3]CMM¹³⁺ always form irrespective of the one-electron reduction potential of the oxidant?

Figure R2.

(3) Regarding to the origin of the clockwise direction selectivity in step 2

In Supplementary Fig. 40, the authors rationally explained the origin of clockwise direction selectivity in step 2. However, when the discussion starting from the high energy state (oxidized state II), in my opinion, the high selectivity for path O1 (clockwise direction) could be difficult to explain. In Supplementary Fig. 40, path O2 has relatively flat potential energy surface around the oxidized state II, which could also drive the path O2 (counterclockwise direction) in some extent. Could you consider (discuss) the possibility that the observed direction selectivity comes from the difference of the one- electron reduction potentials of the two metastable oxidized states resulting from counterclockwise and clockwise direction (Figure R3)? Among the two metastable states, the metastable oxidized state from counterclockwise has less negative one-electron reduction potential due to electrostatic repulsion from the nearby PY+ (Figure R3). Therefore, conversely the metastable oxidized state resulting from counterclockwise is difficult to be formed by the oxidation process of $[3]CMM^{7+} + 6e^-$ (II).

Figure R3.

(4) Regarding to the metastable state at $t = 0$.

In Supplementary Fig. 69, the final oxidized product co-exists even at $t = 0$. Judging from the integration ratio between H26 and H26*, the ratio between the final and metastable oxidized products

seems to be almost 1:1 at $t = 0$. What is the origin of the final oxidized product at $t = 0$? Does it come from the counterclockwise rotation? The metastable oxidized state III has 100% clockwise rotation selectivity, and the authors estimated the probability of clockwise rotation by one redox cycle as 85%. Consequently, the final oxidized product at $t = 0$ should have near 50/50 probability of clockwise and counterclockwise direction. Could the authors determine the rotation probability of the final and metastable oxidized products at $t = 0$ by using $[Dn]-[3]CMM \cdot 13PF_6$ (Phen' and Phen")?

(5) Regarding to the one redox cycle.

In the present system, 180 degree rotation was performed by 6 equiv Cp_2Co , and the re-oxidization by using 6 equiv $NOPF_6$ (which one reduced and oxidized first?). Does the rotation selectivity change when 180 degree rotation is driven by sequential addition of Cp_2Co (2, 4, 6 equiv) and the re-oxidization by $NOPF_6$ (2, 4, 6 equiv)?

(6) Regarding to kinetic cycles of step 1 and 3.

In contrast to the rich insights into the experimental data on the kinetics of step 2, that of step 1 and 3 are not discussed. Could the authors investigate the kinetic cycles of step 1 and 3 by electron spin resonance (ESR) spectroscopy? Is there also metastable species in the reduced state? In this context, the authors successfully analysed the similar radical species by ESR in recent nature paper (Fig. 3e, *Nature* 603, 265–270 (2022)).

(7) Regarding to QM calculations

In Supplementary Fig. 39 and 40, the authors performed the QM calculations by moving the two rings sequentially. This is smart to reduce the calculation complexity in the present system, while in the real system, the two rings move synchronously due to the electrostatic repulsion between the two rings. The present QM calculations are implicitly assumed that the electrostatic repulsion energy between the two rings is constant (or negligible) in the cycles of Supplementary Fig. 39b and 40b. Probably, the total energy (in each point) can be described as $E_{total} = E_A + E_B + E_{AB}$, where E_A and E_B denote the interaction between the loop and the rings (A and B); E_{AB} is the electrostatic repulsion energy between the two rings. Could the authors determine the E_{AB} value in each point in Fig. 39c and 40d by energy decomposition analysis (EDA) of the structure optimized at each point? The contribution of the electrostatic repulsion energy between the two rings on potential energy surface in Fig. 39c and 40d (E_{AB}/E_{total}) would be informative to be considered the geared fashion in this system.

(8) Regarding to Supplementary Fig. 40

Which one corresponds to the metastable state of $[3]CMM^{13+}$ in Supplementary Fig. 40d? Is structure (19,3)? If so, please mention it clearly.

(9) Regarding to assignment of redox peaks in Fig. 3c

There are many redox peaks in the cyclic voltammogram of [3]CMM•13PF₆ (Fig. 3c, (76/-80,-150), (-164/-246), (-687/-768), (-825/-914), (-935/-1010) mV). Therefore, the Supplementary Table for the assignment of each redox peak would be helpful.

(10) Regarding to Supplementary Section 12

Providing a brief discussion on Supplementary Section 12 (Metastable State and Kinetic Studies) would be helpful.

Some trivial points:

Page 5, line (from the top), "Fig 2c" should be "Fig. 2c".

Page S65, line 5 (from the top), "CBPQT^{2(+•)} ring 1" should be "CBPQT⁴⁺ ring 1".

Referee #3 (Remarks to the Author):

In this manuscript, Zhang et al. report an artificial molecular rotary motor that operates continuously under application of an oscillating electric field. Artificial molecular machines provide minimal, analytically tractable models of the fascinating and hugely structurally and functionally complex molecular machinery of biology. These synthetic endeavours will ultimately forge a route to human technologies that exploit artificial dynamic molecular systems driven away from equilibrium by an external power source. This will be quite unlike any existing manmade technology. The challenge of achieving continuous directional motion in abiotic molecular structures however remains considerable, with only a handful of examples previously described.

Recently, there has been a leap forward in the sophistication of artificial unidirectional rotary and linear motors, driven, at least in part, by a new design-led era that is founded on an improved understanding among synthetic chemists of how the principles of ratcheting mechanisms can be applied to arrive at working molecular structures. Continuously rotating light powered[ref. 38] and chemically powered[refs. 14, 36, 41] motors have already been achieved. The current work completes the trilogy by reporting continuous unidirectional rotary motion powered by electricity. There are three key points of novelty:

1. The redox mechanism can be driven continuously for multiple cycles by application of an oscillating electric field, producing zero chemical waste. While the latter feature is common with light-driven motors, harnessing an electrical power source suggests myriad opportunities for

interfacing with other nanoscale electrical technologies.

2. The directionality of motion is an emergent feature of the doubly-interlocked [3]catenane molecular architecture. The [3]catenane is assembled from two identical small rings, each mechanically interlocked around one large ring. The latter can be viewed as a "track" around which the small rings move. [3]Catenane architectures have been used in two earlier pioneering examples of light-fuelled[Leigh 2003, ref. 13] and chemically fuelled[Leigh 2017, ref. 14] rotary motors, but the movement of the current system is different in that it achieves both synchronous and cooperative movement of the two rings.

During one reduction-oxidation cycle, the small rings swap places, each one moving approximately 180° in a direction that is defined by the presence of the other ring. Although this is reminiscent of the earlier light-fuelled design[ref. 13] in that the movement is "cooperative" (the position of one ring affects the movement of the other), in the Leigh system, the rings moved sequentially in a "follow-the-leader" fashion (ie not synchronously). In the previous chemically fuelled design,[ref. 14] the rings do move at the same time (synchronous), but in this case the motion of one ring is not influenced by motion of the other (not cooperative).

3. The operating mechanism does not require either cleavage or isomerisation of any covalent bonds. This is fundamentally different to either light or chemically powered systems.

As an important first in class report, this manuscript will generate considerable interest across broad constituencies of supramolecular and nanochemistry, materials science and biology. At a conceptual level, it is exactly the sort of study appropriate for publication Nature. Strong evidence for the claimed motor operation is provided in the form of both experimental and computational data. However, there are some issues with the analysis and presentation of the evidence and the description of the operating mechanism that first require attention.

1. Ratcheting mechanism

(a) The authors describe the mechanism as combining "energy and information ratchets" (p2, lines18-21; also p3, line 15; p 5, line 9; p9, line 15)

I find this inconsistent with the definitions of these mechanisms that the authors themselves have been at the forefront of coining and translating for consistent and accurate use by the community. Furthermore, this statement is inconsistent with the sentence that follows (P2, lines 21-23) "The two rings take turns at being energy ratcheted and then playing the role of a brake to serve as the barrier by which the other ring is energy ratcheted".

An information ratchet mechanism involves a manipulation of the potential energy surface that differs in rate according to the position of the moving object (the small rings here).[e.g. Astumian, Eur Biophys J, 1998, 27, 474; PhysChemChemPhys 2007, 9, 5067; ACS Nano. 2015, 9, 8672] The information-transfer process involves energy consumption, so that the mechanism does not violate the second law. In a minimal sense, only a position-dependent change in an energy barrier is required. Alternatively, an informational (or "allosteric") control over the manipulation of energy wells depending on the macrocycle position can be an efficiency improving feature of an energy ratchet.[Eur Biophys J 1998; ACS Nano. 2015] So it is certainly the case that a real-world mechanism can involve elements of both the two minimal extremes.

In the current design, the potential energy surface experienced by one ring is defined by the position of the other ring. However, the "information" here is purely encoded in the molecular structure of the motor. There is no energy-consuming information transfer process, nor is there an informational control over the switching. The motor would not operate in the absence of the energetic positional bias that is switched by the electric stimulus.

In all of these regards, the directional motion does not arise through a mechanism which I can

understand as an information ratchet. Indeed, the related light-fuelled [3]catenane rotary motor previously created by Leigh and co-workers[ref. 13] has been described by the current authors as an energy ratchet.[e.g., PhysChemChemPhys 2007; J. Am. Chem. Soc. 2021, 143, 5569] The current motor operates in a very similar fashion and, to my mind, should be described as an energy ratchet. At the very least, the rationale for describing the mechanism as including elements of an information ratchet – and how this is consistent with previous expositions of this concept – needs to be more clearly explained.

(b) Irrespective of the definition of the overall mechanism, it is misleading to describe individual (co-)conformational movements as “energy ratcheted” or “information ratcheted” (p2, lines 22-23 and p5, lines 6-11). “Information” cannot ratchet motion – motion is constrained by an energy barrier, which is a consequence of the molecular structure of the machine (this certainly is a form of “information” but that is a truism of every single molecular structure).

All molecular movements are thermal fluctuations across a potential energy surface. In the current example, each step involves a co-conformational rearrangement from a high-energy metastable state to a lower energy minimum in a direction that is biased by unequal kinetic barriers. The term “biased Brownian motion” has previously been used to describe such steps by the authors,[e.g. Eur Biophys J 1998] and has been adopted by others.[e.g., J. Am. Chem. Soc. 2006, 128, 4058] I would advocate the continued consistent application of this terminology.

2. Evidence for directionality and synchronous macrocycle movement

(a) The DFT studies are beautifully presented (Figures 39 and 40 are particularly informative) and provide very strong support for the proposed sequence of chemomechanical changes. This is important in the context of establishing the operation of such a complicated system, for which any one piece of experimental evidence does not provide incontrovertible proof alone. The excellent agreement between experiment and computation together provides very strong evidence.

(b) The preparation and analysis of the [D16]-isotopologue with in-built bias in the positions of two isotopically differentiable macrocycles is an elegant experiment. However, on its own, this experiment proves only swapping of the position of the rings – not evidence of directionality. The inference of directionality is based on the assumption that “the CBPQT2(+•) ring does not pass over the IPP unit under reducing conditions” (Supplementary Information pS85). Previous results on rotaxane-based pumps containing precisely the same triazole-IPP motif provide the essential experimental basis for this assumption. This should be more prominently described in the main text (as should the significance of refs. 50 and 51), in place of the vague statement “a steric barrier (IPP), is key to inducing unidirectional motion” (p4, line 25).

(c) I also do not follow the logic for combining the assumption above with the NMR results via the probability tree to quantify directionality. If the ring does not pass over the IPP unit in the reduced state then, by the definitions of Suppl. Fig. 55, $1 - \text{pred} = 0$, so $\text{pred} = 1$ (not $0.5 < \text{pred} < 1$). Given that no rings can undergo counterclockwise motion in the first half of the cycle, it is trivial to infer that $\text{pox} = 0.85$ to match with the NMR experiments showing 85% of the rings swap positions.

(d) Inherent in all of this discussion is the assumption that the electron-transfer kinetics are fast in comparison to (co-)conformational motions and do not discriminate between V1/V2 sites (ie one site is not switched before the other and the switching is not significantly affected by position of the macrocycles). It is evident from the cyclic voltammetry experiment (Fig. 3c) that, at least in one direction, the sites are discriminated in terms of reduction potential. Some comment on these assumptions and their experimental basis is required.

(e) Structural characterization of the metastable state observed under oxidizing conditions provides excellent experimental evidence for directional rotation during the second half of one cycle. This is because the alternative motion would mean both rings had moved past their

equilibrium position. Stating this key step in the logic would help the reader quickly interpret the significance of this result (p8, lines 20-22).

Mechanical description of machine operation

(a) "the two rings undergo a 3 geared¹⁻³ synchronous and unidirectional circumrotation" (p9, lines 2-3 and elsewhere)

The analogy with macroscopic gears is neither useful nor accurate. A gearing mechanism facilitates the conversion of angular rotary motion into either linear motion or angular motion of a different frequency or direction. In the current molecular machine, the motion of the two small rings are directly and inextricably entrained. There is no way the design could be adapted to achieve gearing effects.

(b) "The [CBPQT-2]^{2(+•)} ring on the BPM unit must move to the V_{2+•} recognition site, while the [CBPQT-1]^{2(+•)} ring on the V_{1+•} recognition site prevents the backward motion" (p5, lines 6-8) This statement infers that the rings move in a sequential fashion. However, there is no experimental evidence that the barrier for movement of one ring is significantly lower than that for movement of the other. In fact, the DFT results suggest a synchronous relaxation of both rings to their new equilibrium position in both the oxidised and reduced states. Consistent and accurate language should be applied to help the reader retain an accurate impression of the molecular-level processes.

Minor points

1. I found the presentation of the DFT results in Fig. 2b to be almost impossible to interpret without reference to the Suppl. Info. The x and y axes on this plot must be labelled. The authors should consider removing the detailed cross-grid, which is meaningless without the detail provided in the SI; improved explanatory text should be included in the caption.

2. Conversely, Suppl. Figs. 39 and 40 are hugely informative but could be further improved.

(a) Provide the coordinates of the starting/end states (I and II) in Suppl. Fig. 39c and 40d.

(b) Provide an indication of what the minimized structures given in Fig. 39e/f and Fig. 40e/f correspond to (high energy structures representing each kinetic barrier) and provide cross references to where the minimized structures of each low-energy state can be found (i.e. oxidised I / reduced II in Fig. 2a; metastable state in Fig. 5a).

(c) Suppl. Fig. 40b is missing the labels for structural elements (T, PY+, etc) on the x-axis.

3. A consistent numbering system is used to mark locations around the large ring throughout the manuscript. This is very useful for interpreting the DFT results for example. However, arabic numerals (1/2) are also used to differentiate the CBPQT rings and viologen stations. Further, the cartoon diagrams include a numeral to indicate the number of carbon atoms connecting each unit. The manifold use of the same sort of label hampers readability.

4. There is no interpretation or discussion of the cyclic voltammetry results presented in Fig. 3c (either in the main text or Suppl. Info.). Presumably, these results were used to inform the voltages chosen for controlled potential electrolysis experiments. The CV results should also reveal information regarding the rate of electron transfer processes (see comment above). The direction of oxidising/reducing current should be indicated on the y-axis of the cyclic voltammetry plot and each electron transfer process should be labelled.

5. The authors should comment on the choice of voltages used for the controlled potential electrolysis procedure, particularly given that different settings were used in the one cycle and repeated cycle experiments (p S70).

6. The description of the preparation of positionally biased isotopologues as "in a controlled manner" (p7, line 12-15) does not help the reader understand the nature of this experiment. A brief description of the synthetic strategy involving sequential threading would significantly

improve readability of this section.

7. The schematic and photograph of the electrolysis setup (Fig. 4a/b) does not aid understanding. There does not seem to be any significant innovation to this (if there is, it is not described; the same set up appears to have previously been reported in ref. 19). Including this detail in the main manuscript detracts from the novel concepts reported.

8. p9, lines 6-9: "...dramatically different from previously reported [3]catenane-based molecular motors^{13,14} in that our electric molecular motor can operate continuously and synchronously, and is driven by a single energy source."

Achieving continuous and synchronous motion in one system is certainly a novel feature of the current example. However, the previously reported examples each also operate by "a single energy source" (photons in ref. 13 and protons in ref. 14). The differentiating factor is that other stimuli are required to operate these earlier machines. It should be made clear that these reagents do not provide energy that powers the motion.

9. The mass spectrometry results (Suppl. Info. Section 4) require some interpretive explanation.

10. There are some subtle but significant differences in the ¹H NMR spectra presented in Suppl. Fig. 35 (e.g. chemical shifts of the highlighted BPM protons). Similarly for the spectra in Suppl. Fig. 45. The authors should briefly comment on the origin of these differences.

11. The significance or contribution of the data on the relaxation of the deuterated isotopologues presented in Suppl. Figs. 67 and 68 is not clear. This should be described in the Suppl. Info. or else these experiments removed.

Author Rebuttals to Initial Comments:

Listed below are the details of our responses to the referees' comments.

Reviewers' comments

Referee #1:

The authors present a molecular motor based on a [3]catenane architecture. The motor comprises two cyclobis(paraquat-p-phenylene) (CBPQT⁴⁺) rings mechanically interlocked around a track consisting of multiple recognition sites and steric/electronic barriers. The motor employs an energy ratchet design, reminiscent of the strategy employed by Leigh in 2003,¹ where one macrocycle blocks the backwards movement of the other. The system is powered by the Stoddart group's well-established 2013 redox-driven energy ratchet (dubbed a pumping cassette).² The redox ratchet has seen multiple rounds of iteration, having been optimised to its current form in 2015,³ and having been demonstrated to work electrochemically in 2018,⁴ as such, the system is robust and well-understood, and thus provides a good basis for the development of this new motor. As with these previous related systems, successive reduction and oxidation of the CBPQT⁴⁺ rings and viologen binding sites allows directional movement of the rings. The macrocycles preferentially pass over a pyridinium cationic barrier towards the viologen⁺⁺ binding sites with CBPQT²⁽⁺⁾ in the reduced diradical dicationic form, but over a isopropylphenylene steric barrier towards alternative binding sites when reoxidised to CBPQT⁴⁺. As opposed to previous versions, which progressively pump macrocycles out of solution onto a thread,³⁻⁹ or along a linear track,^{2,9,10} this iteration of the ratchet is used to control the rotary motion of the CBPQT⁴⁺ rings around a circular track.

The authors demonstrate the use of controlled potential electrolysis to operate this motor using an electrical current, affording a non-invasive, repeatable and clean (in that no waste products are formed) method for reliable operations. Alternatively, the authors show that the motor can also be successfully operated by a wide array of chemical reductants and oxidants. Finally, the authors investigate the directionality of the motor, using a deuterium labelled CBPQT⁴⁺ macrocycle to establish that the rings swap positions during a redox cycle. 85% of motors complete a 180° clockwise rotation per redox cycle. This conclusion is supported by a series of kinetic models and computational simulations of the potential energy surface.

The structure of the report is logical, the experimental work is well presented and the supplementary information is detailed and justifies their observations. However, I have a few

concerns about the discussion that might be clarified:

1. The authors describe their motor as operating by a combination of an energy ratchet, information ratchet. This seems incorrect to me, and indeed the motor appears to operate purely by an energy ratchet mechanism.¹¹ In the operation cycle of the motor, oxidation or reduction breaks the balance of the system, with the macrocycles placed in a thermodynamically unfavoured state. The macrocycles subsequently relax to a local (kinetically defined) equilibrium position via the most kinetically accessible pathway, constituting a power stroke.

By contrast, in an information ratchet,^{11,12} the key component is the kinetic gating of the chemical, photochemical, or electrochemical reactions that switch the ratchet between states, such that the reaction occurs more rapidly when it enables forward movement (as is well explained in the authors' previous work).¹³

It is possible that some element of information ratchet mechanism may be present if the rate of reduction occurs more rapidly with CBPQT⁴⁺ over the triazole and bis(4-methylene-phenyl)methane stations and/or that the rate of oxidation occurs more rapidly when CBPQT²⁽⁺⁾ binds the viologen⁺ units. This must then result in a kinetic bias for forward movement. No such evidence is currently provided, and I recommend the authors remove this claim unless they can provide such evidence.

Reply: On reflection, we accept that the information ratchet component of the mechanism in the present article is extremely subtle. Consequently, we have deleted mention of the information ratchet in the revised manuscript, since indeed, the energy ratchet mechanism captures adequately the behaviour of the [3]catenane molecular motor.

2. The mechanism of operation of the motor closely reflects Leigh's 2003 [3]catenane¹ (which operates purely by an energy ratchet mechanism). I consider that the authors should explicitly highlight the connection to this previous work, particularly the use of a second macrocycle to block backwards motion of the first.

Reply: The operating mechanism of the electric molecular motor has some similarities with that of the Leigh's 2003 [3]catenane motor, especially regarding the blocking of the backward motion of one ring by the presence of the other ring. Unlike Leigh's motor, however, the electric motor described in our article requires modulation of only a single stimulus — namely the continuous oscillation of redox potential, with the directionality being determined by kinetic asymmetry.

3. The authors could also more explicitly highlight their own papers that this new system builds on,²⁻¹⁰ especially papers detailing the origin of the techniques used for this motor.²⁻⁵

Reply: We have highlighted our previous studies that the [3]catenane molecular motor builds on in the revised manuscript.

4. Feringa has previously demonstrated an electrically-driven molecular motor, which should be cited.¹⁴ The authors might also relate their work to the electrically driven motors of Sykes.¹⁵ In both cases, the authors may wish to highlight the substantial differences between their system and this previous work.

Reply: We have cited both Feringa's and Sykes' electrically driven motors at an appropriate position (Page 3, Line 8) in the revised manuscript. These single-molecule motors, which rotate around a covalent (Feringa) or coordination (Sykes) bond, were driven by using tunneling currents under ultrahigh vacuum on surfaces. Their operations relies on a very different mechanisms from that employed in our research.

5. The authors could emphasise and expand upon the key novel aspects of their system, such as the advantages of the rapid and clean electrochemical operation and good directional fidelity.

Reply: We have replaced the discussion of the information ratchet with an expanded description of the advantages of the operation of a molecular machine electrochemically with no chemical waste produced, and where the directional fidelity is excellent.

6. The analogy between the reported system and molecular gears feels rather forced to me. I'm not sure that it is either relevant nor valid and it could be removed entirely. The term 'Molecular Gears' is generally used to describe intermeshed components, whereby rotation about an axis of one component induces a rotation about the axis for another component. This system does not utilise interaction.¹⁶

Reply: We maintain that the directional motion of our [3]catenane motor is indeed an example of intramolecular gearing. The interactions between the two mobile rings in the electric molecular motor allow organized behaviour under the action of an external stimulus. This emergent behaviour is, to our way of thinking, the essence of molecularly geared motion, wherein the interaction between the two rings, combined with their respective kinetics, gives rise to directional motion.

7. The authors state in the conclusion that the motor operates “continuously”. As the conditions must be switched back and forth between reductive and oxidative, this is misleading as it implies that only a single set of conditions are needed for operation (as for light driven motors^{17,18} or chemically fuelled information ratchet motors^{19,20}). The authors should clarify that the motor operates by continuous oscillation of redox potential.

Reply: The referee is correct in that the motor operates by continuous oscillation of redox potential. The electrically driven [3]catenane motor relies on the external modulation to provide the timing stimulus. We have clarified that the motor operates by the continuous oscillation of redox potential in the revised manuscript.

References

1. Leigh, D. A., Wong, J. K. Y., Dehez, F. & Zerbetto, F. Unidirectional rotation in a mechanically interlocked molecular rotor. *Nature* **424**, 174–179 (2003).
2. Li, H. *et al.* Relative unidirectional translation in an artificial molecular assembly fueled by light. *J. Am. Chem. Soc.* **135**, 18609–18620 (2013).
3. Cheng, C. *et al.* An artificial molecular pump. *Nat. Nanotechnol.* **10**, 547–553 (2015).
4. Pezzato, C. *et al.* Controlling dual molecular pumps electrochemically. *Angew. Chem. Int. Ed.* **57**, 9325–9329 (2018).
5. Qiu, Y. *et al.* A precise polyrotaxane synthesizer. *Science* **368**, 1247–1253 (2020).
6. Feng, L. *et al.* Active mechanisorption driven by pumping cassettes. *Science* **374**, 1215–1221 (2021).
7. Li, X. *et al.* Fluorescence quenching by redox molecular pumping. *J. Am. Chem. Soc.* **144**, 3572–3579 (2022).
8. Cai, K. *et al.* Molecular-pump-enabled synthesis of a daisy chain polymer. *J. Am. Chem. Soc.* **142**, 10308–10313 (2020).

9. Guo, Q.-H. *et al.* Artificial molecular pump operating in response to electricity and light. *J. Am. Chem. Soc.* **142**, 14443–14449 (2020).
10. Qiu, Y. *et al.* A molecular dual pump. *J. Am. Chem. Soc.* **141**, 17472–17476 (2019).
11. Kay, E. R., Leigh, D. A. & Zerbetto, F. Synthetic molecular motors and mechanical machines. *Angew. Chem. Int. Ed.* **46**, 72–191 (2007).
12. Serreli, V., Lee, C.-F., Kay, E. R. & Leigh, D. A. A molecular information ratchet. *Nature* **445**, 523–527 (2007).
13. Astumian, R. D. Kinetic asymmetry allows macromolecular catalysts to drive an information ratchet. *Nat. Commun.* **10**, 3837 (2019).
14. Kudernac, T. *et al.* Electrically driven directional motion of a four-wheeled molecule on a metal surface. *Nature* **479**, 208–211 (2011).
15. Tierney, H. L. *et al.* Experimental demonstration of a single-molecule electric motor. *Nat. Nanotechnol.* **6**, 625–629 (2011).
16. Gisbert, Y., Abid, S., Kammerer, C. & Rapenne, G. Molecular gears: from solution to surfaces. *Chem. Eur. J.* **27**, 12019–12031 (2021).
17. Pooler, D. R. S., Lubbe, A. S., Crespi, S. & Feringa, B. L. Designing light-driven rotary molecular motors. *Chem. Sci.* **12**, 14964–14986 (2021).
18. Koumura, N., Zijlstra, R. W. J., van Delden, R. A., Harada, N. & Feringa, B. L. Light-driven monidirectional molecular rotor. *Nature* **401**, 152–155 (1999).
19. Wilson, M. R., Solà, J., Carlone, A., Goldup, S. M., Lebrasseur, N. & Leigh, D. A. An autonomous chemically fuelled small-molecule motor. *Nature* **534**, 235–240 (2016).
20. Borsley, S., Kreidt, E., Leigh, D. A. & Roberts, B. M. W. Autonomous fuelled directional rotation about a covalent single bond. *Nature* **604**, 80–85 (2022).

Referee #2:

Stoddart et al reported an electric molecular motor with unidirectional rotary motion. First of all, this reviewer is convincing the authors' conclusions that this molecular motor exhibits the unidirectional movement driven by the redox cycles. The experiments with using $[D_n]$ - $[3]CMM \cdot 13PF_6$ (Fig. 4e) elegantly demonstrates the unidirectional rotary motion (85% probability for clockwise rotation). All experiments were well-designed and performed with high degree of perfection (including reference citations). Thus, the scientific value of this contribution is worth to be published in Nature. The authors are advised to address the following suggestions and questions, if the suggested discussions provide the additional scientific value in this significant work.

(1) Regarding to rotation of D-labeled CBPQT⁴⁺ upon more than two redox cycles:

In Supplementary Fig. 45, the authors successfully provided ¹H NMR spectrum of $[3]CMM \cdot 13PF_6$ after five redox cycles, and confirmed the good reversibility of the rotation of CBPQT⁴⁺ rings. Conversely, in the case of the D-labeled $[3]CMM \cdot 13PF_6$, ¹H NMR spectra only after one redox cycles (corresponding to 180 degree rotation) were given in Fig. 4e and Supplementary Figs. 52 and 53. Could the authors provide ¹H NMR spectra of $[D_n]$ - $[3]CMM \cdot 13PF_6$ after more than two redox cycles? I would like to see the positional exchange between Phen' and Phen'' after more than the 360 degree rotation. The ratio between Phen' (deuterated-poor ring) and Phen'' (deuterated-rich ring) would be changed depending on the number of redox cycle. Here, I shall distinguish the two CBPQT⁴⁺ rings as rings A and B, respectively. In order to aid the discussion, the rotation is started from [ring A]:[ring B] = 100:0 and 0:100 at V1²⁺ and V2²⁺, respectively (Figure R1). In accordance to the 85% probability for clockwise rotation, the ratio between rings A and B can change depending on the number of redox cycle to approach [ring A]:[ring B] = 50:50. Accordingly, the electric molecular motor can record history of the number of rotations in statistical meaning (using huge number of molecules). This could not be achieved by the macroscopic motor (normally used solely) with perfect unidirectional rotation. This is, for me, interesting.

Figure R1.

Reply: We thank the referee for these observations. The referee explains correctly how the ratio between rings A and B can be changed depending on the number of redox cycles in the Figure R1. As the referee suggested, we have added the ^1H NMR spectra of $[\text{D}_n]\text{-}[\mathbf{3}]\text{CMM}\cdot 13\text{PF}_6$ after three redox cycles in the revised Supplementary Information. The spectra are also illustrated below.

Supplementary Fig. 61 | Partial ^1H NMR spectra of $[\text{D}_n]\text{-}[\mathbf{3}]\text{CMM}\cdot 13\text{PF}_6$. (I) Initial spectrum; (II) / (III) / (IV) after 1 / 2 / 3 redox cycles of a chemically driven operation by using Zn dust and NOPF_6

Theoretically, it is true that the electric molecular motor can record history of the number of rotations in a statistical manner: this point is a very interesting one. As the ratio between rings A and B becomes close to 50 : 50 after multiple redox cycles, it will be more difficult to obtain the correct ratio experimentally. The accuracy possible from integration is limited by the resolution that can be achieved by ^1H NMR spectroscopy.

(2) Regarding to the electron-transfer mechanism (origin of metastable state of [3]CMM¹³⁺)

Most probably, the authors consider the electron transfer process like vertical transition as shown in Figure R2 (mechanism A), where therefore the energetically unstable species with high electrostatic repulsion (oxidized state II in Supplementary Fig. 40a) was assumed. In accordance with the theory (while formally 6e⁻ electron transfer), electron transfer coupled with partial rotation of the two rings would be more straightforward (mechanism B in Figure R2), where the reduced- and oxidized-states have the same reaction coordinate at the closing point (reduced state X and oxidized state X). I understand that this point has no huge impact on the conclusion of unidirectional rotation. However, the electron-transfer mechanism may be important for interpreting the origin of metastable state [3]CMM¹³⁺ (: when the two rings start to move). Please consider (discuss) the possibility of mechanism B, and tell me the reason which is more convincing mechanism in this case. In addition, does the metastable state [3]CMM¹³⁺ always form irrespective of the one-electron reduction of the oxidant?

Figure R2.

Reply: We thank the referee for asking this important question. The question is related to our asking whether ligand-induced conformational changes occur by the so-called induced fit related to mechanism A or the preexisting equilibrium related to mechanism B. In our case, we believe

the strength of radical-pairing interaction ($\Delta G = -6.2 \text{ kcal mol}^{-1}$, *Nat. Rev. Chem.* **2021**, 5, 447–465) suggests that the contribution of mechanism B to the overall behaviour is relatively small. Furthermore, the experimental conditions we used are such that the oxidation to the oxidized state $[3]\text{CMM}^{13+}$ is both complete and very rapidly.

(3) Regarding to the origin of the clockwise direction selectivity in step 2

In Supplementary Fig. 40, the authors rationally explained the origin of clockwise direction selectivity in step 2. However, when the discussion starting from the high energy state (oxidized state II), in my opinion, the high selectivity for path O1 (clockwise direction) could be difficult to explain. In Supplementary Fig. 40, path O2 has relatively flat potential energy surface around the oxidized state II, which could also drive the path O2 (counterclockwise direction) in some extent. Could you consider (discuss) the possibility that the observed direction selectivity comes from the difference of the one-electron reduction potentials of the two metastable oxidized states resulting from counterclockwise and clockwise direction (Figure R3)?

Among the two metastable states, the metastable oxidized state from counterclockwise has less negative one-electron reduction potential due to electrostatic repulsion from the nearby PY^+ (Figure R3). Therefore, conversely the metastable oxidized state resulting from counterclockwise is difficult to be formed by the oxidation process of $[3]\text{CMM}^{7+6\cdot}$ (II).

Figure R3.

Reply: We thank the referee for posing this question. Note (Supplementary Fig. 40d) the rapid jump at II (28, 10) upon oxidation is a result of the relative speed of oxidation compared with ring motion. Once the oxidation has occurred, there is a high barrier to counterclockwise motion. The guiding principle is that the CBPQT²⁽⁺⁾ ring almost never passes over an IPP unit, and the CBPQT⁴⁺ ring almost never passes over a PY⁺ unit. These experimentally based facts raise the question posed by the referee — what happens if some, but not all, of the viologen units are reduced/oxidized at any one time? This question is the grist for subsequent study. We have targeted the present research on situations where reductions and oxidations are accomplished very rapidly and completely. The ring motions occur following the reduction and oxidation. The situation where not all of the viologen units are in the same redox state is beyond the scope of this study.

(4) Regarding to the metastable state at t = 0.

In Supplementary Fig. 69, the final oxidized product co-exists even at t = 0. Judging from the integration ratio between H26 and H26*, the ratio between the final and metastable oxidized products seems to be almost 1:1 at t = 0. What is the origin of the final oxidized product at t = 0? Does it come from the counterclockwise rotation? The metastable oxidized state III has 100% clockwise rotation selectivity, and the authors estimated the probability of clockwise rotation by one redox cycle as 85%. Consequently, the final oxidized product at t = 0 should have near 50/50 probability of clockwise and counterclockwise direction. Could the authors determine the rotation probability of the final and metastable oxidized products at t = 0 by using [D_n]-[3]CMM•13PF₆ (Phen' and Phen'')?

Reply: In the kinetic studies, we define the starting time t = 0 as the time when we record the first ¹H NMR spectrum. Considering the fact that it takes some time to prepare the sample and obtain the first ¹H NMR spectrum, some of the molecules in the metastable state have already relaxed to the final stable oxidized state. It is for this reason that the ratio between the final and metastable oxidized products is almost 1:1 in the first ¹H NMR spectrum. In an analogous experiment (Supplementary Fig. 72 in the revised Supplementary Information) performed at 273 K, the ratio between the metastable and final oxidized state is closed to 2:1. We believe that the selective clockwise rotation from the metastable oxidized state III is 85% (not 100%) and that the remaining 15% of the metastable state III takes less-kinetically favoured pathways to the thermodynamic minima, which are not easy to observe by ¹H NMR spectroscopy.

(5) Regarding to the one redox cycle.

In the present system, 180 degree rotation was performed by 6 equiv Cp_2Co , and the re-oxidization by using 6 equiv NOPF_6 (which one reduced and oxidized first?). Does the rotation selectivity change when 180 degree rotation is driven by sequential addition of Cp_2Co (2, 4, 6 equiv) and the re-oxidization by NOPF_6 (2, 4, 6 equiv)?

Reply: In our experiments, all of the reductions and oxidations occur rapidly relative to the movement of the rings. As a result, the order of reduction and oxidation of the different viologen units is inconsequential as far as the circumrotation of the rings in the [3]catenane motor are concerned. The answer to the specific questions posed by the referee depends on the timing of additions and the kinetics of the system. If the reductions and oxidations are very slow, the system will remain in equilibrium at every instant and no directional rotation would result from this modulation of a single parameter (redox potential).

(6) Regarding to kinetic cycles of step 1 and 3.

In contrast to the rich insights into the experimental data on the kinetics of step 2, that of step 1 and 3 are not discussed. Could the authors investigate the kinetic cycles of step 1 and 3 by electron spin resonance (ESR) spectroscopy? Is there also metastable species in the reduced state? In this context, the authors successfully analysed the similar radical species by ESR in recent nature paper (Fig. 3e, Nature 603, 265–270 (2022)).

Reply: In comparison with the oxidation step 2 (especially for the thermal relaxation process), the reduction step is a much faster process, an observation which is implicit from the reaction described in the main text, “*The reduced state [3]CMM^{7+6•} of the [3]catenane, which was produced on the addition of 6.0 equivalents of cobaltocene³¹ (Cp_2Co), or an excess of Zn dust³⁰, into an MeCN solution of [3]CMM•13PF₆ was accompanied by an instantaneous change from colourless to dark purple.*”. There are no metastable states in step 1 with a comparable life-time to that in step 2. According to the calculation (Supplementary Table 5 and Supplementary Fig. 39c) for the reduced state [3]CMM^{7+6•}, the Coulombic barrier is calculated to be 13.0 kcal mol⁻¹, corresponding to an average rate constant k of $1.8 \times 10^3 \text{ s}^{-1}$. The half-life of this first-order reaction could be estimated to be 0.4 ms. On such a short time-scale, it is very difficult to investigate the kinetics of step 1 and observe directly the metastable species in the reduced state.

(7) Regarding to QM calculations

In Supplementary Fig. 39 and 40, the authors performed the QM calculations by moving the two rings sequentially. This is smart to reduce the calculation complexity in the present system, while in the real system, the two rings move synchronously due to the electrostatic repulsion between the two rings. The present QM calculations are implicitly assumed that the electrostatic repulsion energy between the two rings is constant (or negligible) in the cycles of Supplementary Fig. 39b and 40b. Probably, the total energy (in each point) can be described as $E_{\text{total}} = E_A + E_B + E_{AB}$, where E_A and E_B denote the interaction between the loop and the rings (A and B); E_{AB} is the electrostatic repulsion energy between the two rings. Could the authors determine the E_{AB} value in each point in Fig. 39c and 40d by energy decomposition analysis (EDA) of the structure optimized at each point? The contribution of the electrostatic repulsion energy between the two rings on potential energy surface in Fig. 39c and 40d (E_{AB}/E_{total}) would be informative to be considered the geared fashion in this system.

Reply: Before we calculated the potential energies of the [3]catenane [3]CMM¹³⁺, we also hypothesized that the total energy (in each point) can be described as $E_{\text{total}} = E_A + E_B + E_{AB}$, where E_A and E_B denote the interaction between the loop and the rings (A and B) and E_{AB} is the electrostatic repulsion energy between the two rings in solution. Unexpectedly, the calculations reveal that this simple energy expression only captures the energy landscape with an accuracy of ~20 kcal/mol. Here we chose to examine the potential energies of the [3]catenane [3]CMM¹³⁺, because the electrostatic repulsion between the two rings in the oxidized state [3]CMM¹³⁺ is much stronger than that in the case of the reduced state [3]CMM^{7+6•}.

We can obtain E_A and E_B for the [3]catenane [3]CMM¹³⁺ by removing one of the CBPQT⁴⁺ rings (B or A) completely and recalculating the potential energy. By plotting (Supplementary Fig. 42) the difference between the E_{actual} and E_A+E_B , which is E_{AB} by definition, versus R_{AB} (defined by the distance between the centers of A and B), we found that the relation is complicated.

Supplementary Fig. 41 | The energies from the actual calculation (E_{actual}) versus the energies from the summation (E_A+E_B) following paths O1 and O2

Supplementary Fig. 42 | The electrostatic repulsion energy E_{AB} between the two CBPQT⁴⁺ rings versus the distance R_{AB} between the centers of the two rings A and B following paths O1 and O2

Supplementary Fig. 43 | The calculated potential energy difference $\Delta E_{A/B}$ between the [3]catenane [3]CMM¹³⁺ and the [2]catenane [2]C⁹⁺ following paths O1 and O2

By plotting (Supplementary Fig. 43) the energy difference $\Delta E_{A/B}$ between the [3]catenane [3]CMM¹³⁺ and the [2]catenane [2]C⁹⁺, we found that the E_A and E_B derived from the [3]catenane [3]CMM¹³⁺ deviates significantly from that for the [2]catenane [2]C⁹⁺ in certain regions — particularly the flexible alkyl chain and the hinge atoms between rigid fragments. This realization indicates that the presence of the other CBPQT⁴⁺ ring in these regions greatly perturbs the geometry of the [2]catenane by hindering the relaxation. This effect is at least as important as the electrostatic repulsion between two CBPQT⁴⁺ rings but is not included in the model of the [2]catenane.

We have included this important discussion in **Section 6.2** of the revised Supplementary Information.

(8) Regarding to Supplementary Fig. 40

Which one corresponds to the metastable state of [3]CMM¹³⁺ in Supplementary Fig. 40d? Is structure (19,3)? If so, please mention it clearly.

Reply: It is structure (19, 3) that corresponds to the metastable state of [3]CMM¹³⁺. We have added this information to Supplementary Fig. 40d in the revised Supplementary Information.

(9) Regarding to assignment of redox peaks in Fig. 3c

There are many redox peaks in the cyclic voltammogram of [3]CMM•13PF₆ (Fig. 3c, (76/-80,-150), (-164/-246), (-687/-768), (-825/-914), (-935/-1010) mV). Therefore, the Supplementary Table for the assignment of each redox peak would be helpful.

Reply: A new section (**Section 8. Cyclic Voltammetry**) including a discussion of the CV results has been added to the revised Supplementary Information.

(10) Regarding to Supplementary Section 12

Providing a brief discussion on Supplementary Section 12 (Metastable State and Kinetic Studies) would be helpful.

Reply: A discussion on the metastable state and kinetic studies has been included in the revised Supplementary Information.

Some trivial points:

Page 5, line (from the top), “Fig 2c” should be “Fig. 2c”.

Page S65, line 5 (from the top), “CBPQT²⁽⁺⁾ ring 1” should be “CBPQT⁴⁺ ring 1”.

Reply: The “Fig 2c” and “CBPQT²⁽⁺⁾ ring 1” have been changed to “Fig. 2c” and “CBPQT⁴⁺ ring A”, respectively.

Referee #3:

In this manuscript, Zhang et al. report an artificial molecular rotary motor that operates continuously under application of an oscillating electric field. Artificial molecular machines provide minimal, analytically tractable models of the fascinating and hugely structurally and functionally complex molecular machinery of biology. These synthetic endeavours will ultimately forge a route to human technologies that exploit artificial dynamic molecular systems driven away from equilibrium by an external power source. This will be quite unlike any existing manmade technology. The challenge of achieving continuous directional motion in abiotic molecular structures however remains considerable, with only a handful of examples previously described.

Recently, there has been a leap forward in the sophistication of artificial unidirectional rotary and linear motors, driven, at least in part, by a new design-led era that is founded on an improved understanding among synthetic chemists of how the principles of ratcheting mechanisms can be applied to arrive at working molecular structures. Continuously rotating light powered[ref. 38] and chemically powered[refs. 14, 36, 41] motors have already been achieved. The current work completes the trilogy by reporting continuous unidirectional rotary motion powered by electricity. There are three key points of novelty:

1. The redox mechanism can be driven continuously for multiple cycles by application of an oscillating electric field, producing zero chemical waste. While the latter feature is common with light-driven motors, harnessing an electrical power source suggests myriad opportunities for interfacing with other nanoscale electrical technologies.
2. The directionality of motion is an emergent feature of the doubly-interlocked [3]catenane molecular architecture. The [3]catenane is assembled from two identical small rings, each mechanically interlocked around one large ring. The latter can be viewed as a “track” around which the small rings move. [3]Catenane architectures have been used in two earlier pioneering examples of light-fuelled[Leigh 2003, ref. 13] and chemically fuelled[Leigh 2017, ref. 14] rotary motors, but the movement of the current system is different in that it achieves both synchronous and cooperative movement of the two rings. During one reduction–oxidation cycle, the small rings swap places, each one moving approximately 180° in a direction that is defined by the presence of the other ring. Although this is reminiscent of the earlier light-fuelled design[ref. 13] in that the movement is “cooperative” (the position of one ring affects the movement of the other),

in the Leigh system, the rings moved sequentially in a “follow-the-leader” fashion (ie not synchronously). In the previous chemically fuelled design,[ref. 14] the rings do move at the same time (synchronous), but in this case the motion of one ring is not influenced by motion of the other (not cooperative).

3. The operating mechanism does not require either cleavage or isomerisation of any covalent bonds. This is fundamentally different to either light or chemically powered systems.

As an important first in class report, this manuscript will generate considerable interest across broad constituencies of supramolecular and nanochemistry, materials science and biology. At a conceptual level, it is exactly the sort of study appropriate for publication Nature. Strong evidence for the claimed motor operation is provided in the form of both experimental and computational data. However, there are some issues with the analysis and presentation of the evidence and the description of the operating mechanism that first require attention.

1. Ratcheting mechanism

The authors describe the mechanism as combining “energy and information ratchets” (p2, lines18-21; also p3, line 15; p 5, line 9; p9, line 15)

I find this inconsistent with the definitions of these mechanisms that the authors themselves have been at the forefront of coining and translating for consistent and accurate use by the community. Furthermore, this statement is inconsistent with the sentence that follows (P2, lines 21-23) “The two rings take turns at being energy ratcheted and then playing the role of a brake to serve as the barrier by which the other ring is energy ratcheted”.

An information ratchet mechanism involves a manipulation of the potential energy surface that differs in rate according to the position of the moving object (the small rings here).[e.g. Astumian, Eur Biophys J, 1998, 27, 474; PhysChemChemPhys 2007, 9, 5067; ACS Nano. 2015, 9, 8672] The information-transfer process involves energy consumption, so that the mechanism does not violate the second law. In a minimal sense, only a position-dependent change in an energy barrier is required. Alternatively, an informational (or “allosteric”) control over the manipulation of energy wells depending on the macrocycle position can be an efficiency improving feature of an energy ratchet.[Eur Biophys J 1998; ACS Nano. 2015] So it is certainly the case that a real-world mechanism can involve elements of both the two minimal extremes.

In the current design, the potential energy surface experienced by one ring is defined by the position of the other ring. However, the “information” here is purely encoded in the molecular structure of the motor. There is no energy-consuming information transfer process, nor is there an informational control over the switching. The motor would not operate in the absence of the energetic positional bias that is switched by the electric stimulus.

In all of these regards, the directional motion does not arise through a mechanism which I can understand as an information ratchet. Indeed, the related light-fuelled [3]catenane rotary motor previously created by Leigh and co-workers[ref. 13] has been described by the current authors as an energy ratchet.[e.g., PhysChemChemPhys 2007; J. Am. Chem. Soc. 2021, 143, 5569] The current motor operates in a very similar fashion and, to my mind, should be described as an energy ratchet. At the very least, the rationale for describing the mechanism as including elements of an information ratchet – and how this is consistent with previous expositions of this concept – needs to be more clearly explained.

Reply: We have removed mention of the information ratchet in the present paper, since indeed, the energy ratchet mechanism captures adequately the behaviour of the [3]catenane molecular motor.

(b) Irrespective of the definition of the overall mechanism, it is misleading to describe individual (co-)conformational movements as “energy ratcheted” or “information ratcheted” (p2, lines 22-23 and p5, lines 6-11). “Information” cannot ratchet motion – motion is constrained by an energy barrier, which is a consequence of the molecular structure of the machine (this certainly is a form of “information” but that is a truism of every single molecular structure).

All molecular movements are thermal fluctuations across a potential energy surface. In the current example, each step involves a co-conformational rearrangement from a high-energy metastable state to a lower energy minimum in a direction that is biased by unequal kinetic barriers. The term “biased Brownian motion” has previously been used to describe such steps by the authors,[e.g. Eur Biophys J 1998] and has been adopted by others.[e.g., J. Am. Chem. Soc. 2006, 128, 4058] I would advocate the continued consistent application of this terminology.

Reply: We have removed the terms “energy ratcheted” and “information ratcheted” in our revised manuscript. As suggested by the referee, we use the term “biased Brownian motion” to describe the steps.

2. Evidence for directionality and synchronous macrocycle movement

(a) The DFT studies are beautifully presented (Figures 39 and 40 are particularly informative) and provide very strong support for the proposed sequence of chemomechanical changes. This is important in the context of establishing the operation of such a complicated system, for which any one piece of experimental evidence does not provide incontrovertible proof alone. The excellent agreement between experiment and computation together provides very strong evidence.

Reply: We thank the referee for the supportive comments on our computational and experimental results.

(b) The preparation and analysis of the [D₁₆]-isotopologue with in-built bias in the positions of two isotopically differentiable macrocycles is an elegant experiment. However, on its own, this experiment proves only swapping of the position of the rings – not evidence of directionality. The inference of directionality is based on the assumption that “the CBPQT²⁽⁺⁾ ring does not pass over the IPP unit under reducing conditions” (Supplementary Information pS85). Previous results on rotaxane-based pumps containing precisely the same triazole-IPP motif provide the essential experimental basis for this assumption. This should be more prominently described in the main text (as should the significance of refs. 50 and 51), in place of the vague statement “a steric barrier (IPP), is key to inducing unidirectional motion” (p4, line 25).

Reply: We agree with the referee that the deuterium experiment alone is not evidence for directionality. The inference of directionality is based on the assumption that “the CBPQT²⁽⁺⁾ ring does not pass over the IPP unit under reducing conditions”. Indeed, we have discussed this latter point in the main text, “*The height of the steric barrier imposed by the IPP unit is independent of the redox state and is large enough ($\Delta E > 25 \text{ kcal mol}^{-1}$) to preclude (Supplementary Figs. 37 and 38) the passage of the CBPQT^{4+/2(+)} ring over the IPP unit.*” Regarding the vague statement “a steric barrier (IPP), is key to inducing unidirectional motion”, the full sentence now reads “*This combination (Fig. 2a) of two mechanically interlocked CBPQT^{4+/2(+)} rings, a switchable barrier, associated with the PY⁺ unit, two switchable viologen (V^{2+/+}) recognition sites, and a steric barrier (IPP), is key to inducing unidirectional motion (Fig. 2b) of the two rings with respect to the loop.*” The relevant elements are controllable barriers and switchable recognition sites combined with the correlated (gearing) motion of several mobile rings to achieved directional motion. As suggested by the referee, we have added references to previous research.

(c) I also do not follow the logic for combining the assumption above with the NMR results via the probability tree to quantify directionality. If the ring does not pass over the IPP unit in the reduced state then, by the definitions of Suppl. Fig. 55, $1 - p_{\text{red}} = 0$, so $p_{\text{red}} = 1$ (not $0.5 < p_{\text{red}} < 1$). Given that no rings can undergo counterclockwise motion in the first half of the cycle, it is trivial to infer that $p_{\text{ox}} = 0.85$ to match with the NMR experiments showing 85% of the rings swap positions.

Reply: According to our calculations and previous investigations, the probability for a CBPQT²⁽⁺⁾ ring to pass over the IPP unit is close to 0, i.e., $1 - p_{\text{red}} \approx 0$. When calculating the directionality, we did not assume the $p_{\text{red}} = 1$ since there is no direct evidence that the ring does not pass over the IPP unit under reducing conditions: we set a reasonable restriction for p_{red} ($0.5 < p_{\text{red}} \leq 1$). The graph shows the mathematic relationships of $p_{\text{ox}}/p_{\text{red}}$ (purple traces) and $p_{\text{ox}}/p_{\text{red}}p_{\text{ox}}$ (solid green traces). If we assume the reliable interval is when $0.85 \leq p_{\text{ox}} \leq 1$ and $0.85 \leq p_{\text{red}} \leq 1$, then we can conclude that $p_{\text{red}}p_{\text{ox}} \approx 0.85$. These calculations based on the observed switching probability provide strong evidence for the unidirectional motion and quantify the directionality to be 85% after one redox cycle.

Supplementary Fig. 64 | Mathematic relationships of $p_{\text{ox}}/p_{\text{red}}$ (purple traces) and $p_{\text{ox}}/p_{\text{red}}p_{\text{ox}}$ (solid green traces)

We have included this Figure in the revised Supplementary Information as Supplementary Fig. 64.

(d) Inherent in all of this discussion is the assumption that the electron-transfer kinetics are fast in comparison to (co-)conformational motions and do not discriminate between V1/V2 sites (ie one site is not switched before the other and the switching is not significantly affected by position of the macrocycles). It is evident from the cyclic voltammetry experiment (Fig. 3c) that, at least in one direction, the sites are discriminated in terms of reduction potential. Some comment on these assumptions and their experimental basis is required.

Reply: In our experiments, all of the reductions and oxidations occur rapidly relative to the ring motion. As a result, the order of reduction and oxidation of the different viologen units is inconsequential for the circumrotation of the rings in the [3]catenane motor. It is indeed assumed that ring motions are much slower than all electronic processes. See the CV data (Supplementary Fig. 47), e.g., when the scan-rate was increased to $2.0 \text{ V}\cdot\text{s}^{-1}$, only one reduction potential peak for the radical state $[\mathbf{3}]\text{CMM}^{7+6\bullet}$ was observed.

(e) Structural characterization of the metastable state observed under oxidizing conditions provides excellent experimental evidence for directional rotation during the second half of one cycle. This is because the alternative motion would mean both rings had moved past their equilibrium position. Stating this key step in the logic would help the reader quickly interpret the significance of this result (p8, lines 20-22).

Reply: In order to go from (Fig. 2c) the metastable state III to the stable state I in the counterclockwise direction would require the two CBPQT⁴⁺ rings to pass over the V²⁺ units, which is obviously energetically unfavourable. We have added a statement to this effect in the main text.

Mechanical description of machine operation

(a) “the two rings undergo a 3 geared 1–3 synchronous and unidirectional circumrotation” (p9, lines 2-3 and elsewhere). The analogy with macroscopic gears is neither useful nor accurate. A gearing mechanism facilitates the conversion of angular rotary motion into either linear motion or angular motion of a different frequency or direction. In the current molecular machine, the motion of the two small rings are directly and inextricably entrained. There is no way the design could be adapted to achieve gearing effects.

Reply: We use the general description of gearing as being a structure by which two degrees of freedom are coupled. In a macroscopic gear, the angle variable for one “cog” would absolutely determine the angle variable for the other “cog”, and the two cogs would, of necessity, move in concert. In contrast, the motion of the “cogs” in molecular gears is softer. A transition of one cog from one angle to the other increases the likelihood of motion of the other cog, but does not “force” its motion in the way we understand in the macroscopic world. The effect is shown elegantly in the two-dimensional energy landscape shown below.

The free energy map of F₀F₁ ATPase

Two-dimensional position map of the [3]catenane molecular motor

Motion along the y-coordinate is certainly influenced by the position along the x-coordinate, and vice-versa, but the motion is accomplished in steps involving motion predominately in the x direction and steps involving motion predominately in the y direction. Although this energy surface happens to have been calculated for the F_0F_1 ATPase by Warshel, the general idea is exactly that illustrated by the computational studies of the [3]catenane in the present paper. The unidirectionality of the F_0F_1 ATPase arises because of the chemical potential difference of ATP and ADP, entering the blue channel from the top more likely than from the bottom, i.e., the unidirectional motion occurs by mass action. The unidirectionality of circumrotation by the two rings in the [3]catenane occurs because of the sequential pulses of reductant and oxidant, but the gearing — the sculpting of the energy landscape that gives rise to a single reactive channel — arises from the design of the [3]catenane molecular motor.

(b) “The [CBPQT-2]²⁽⁺⁾ ring on the BPM unit must move to the V2⁺ recognition site, while the [CBPQT-1]²⁽⁺⁾ ring on the V1⁺ recognition site prevents the backward motion” (p5, lines 6-8). This statement infers that the rings move in a sequential fashion. However, there is no experimental evidence that the barrier for movement of one ring is significantly lower than that for movement of the other. In fact, the DFT results suggest a synchronous relaxation of both rings to their new equilibrium position in both the oxidized and reduced states. Consistent and accurate language should be applied to help the reader retain an accurate impression of the molecular-level processes.

Reply: The referee is correct in that there is no experimental evidence that the barrier for movement of one ring is significantly lower than that for movement of the other. At present, it is not clear whether the motion of rings occur sequentially or synchronously in the reduction process: it is, however, likely that, in the oxidized state, the motion of the rings in transitioning from the metastable state to the final state is concerted. We have removed the sentence “The [CBPQT-2]²⁽⁺⁾ ring on the BPM unit must move to the V2⁺ recognition site, while the [CBPQT-1]²⁽⁺⁾ ring on the V1⁺ recognition site prevents the backward motion” and rewritten this discussion in the revised manuscript.

Minor points

1. I found the presentation of the DFT results in Fig. 2b to be almost impossible to interpret without reference to the Suppl. Info. The x and y axes on this plot must be labelled. The authors should consider removing the detailed cross-grid, which is meaningless without the detail provided in the SI; improved explanatory text should be included in the caption.

Reply: As suggested by the referee, we have labelled the x and y axes and removed the detailed cross-grid. The caption for Fig. 2b has also been modified accordingly.

2. Conversely, Suppl. Figs. 39 and 40 are hugely informative but could be further improved. (a) Provide the coordinates of the starting/end states (I and II) in Suppl. Fig. 39c and 40d. (b) Provide an indication of what the minimized structures given in Fig. 39e/f and Fig. 40e/f correspond to (high energy structures representing each kinetic barrier) and provide cross references to where the minimized structures of each low-energy state can be found (i.e. oxidised I / reduced II in Fig. 2a; metastable state in Fig. 5a). (c) Suppl. Fig. 40b is missing the labels for structural elements (T, PY⁺, etc) on the x-axis.

Reply: As suggested by the referee, (a) the coordinates of the starting/end states (I and II) have been added in the revised Supplementary Fig. 39c and 40d. (b) The descriptions of the minimized structures have been presented in the revised Supplementary Fig. 39 e/f and Fig. 40 e/f, and the cross references for the low-energy state have been added to the captions for the revised Supplementary Fig. 39 and Fig. 40, respectively. (c) The missing labels for structural elements on the x-axis have been added to the revised Supplementary Fig. 40b.

3. A consistent numbering system is used to mark locations around the large ring throughout the manuscript. This is very useful for interpreting the DFT results for example. However, arabic numerals (1/2) are also used to differentiate the CBPQT rings and viologen stations. Further, the cartoon diagrams include a numeral to indicate the number of carbon atoms connecting each unit. The manifold use of the same sort of label hampers readability.

Reply: We have now used A/B to differentiate the CBPQT^{4+/2(+)} rings and removed the labels V1²⁺ and V2²⁺ in the revised manuscript.

4. There is no interpretation or discussion of the cyclic voltammetry results presented in Fig. 3c (either in the main text or Suppl. Info.). Presumably, these results were used to inform the voltages chosen for controlled potential electrolysis experiments. The CV results should also reveal information regarding the rate of electron transfer processes (see comment above). The direction of oxidising/reducing current should be indicated on the y-axis of the cyclic voltammetry plot and each electron transfer process should be labelled.

Reply: The cyclic voltammetry results were used to select the voltages chosen for controlled potential electrolysis experiments. In the revised manuscript, we have combined Fig. 3 and Fig. 4 into a new Fig. 3, in which the CV results have been removed. Accordingly, a new section (**Section 8. Cyclic Voltammetry**) including a discussion of electron transfer processes has been added in the revised Supplementary Information.

5. The authors should comment on the choice of voltages used for the controlled potential electrolysis procedure, particularly given that different settings were used in the one cycle and repeated cycle experiments (p S70).

Reply: The controlled potential electrolysis procedure we used was established in our previous study on Controlling Dual Molecular Pumps Electrochemically in 2018 (*Angew. Chem. Int. Ed.* **2018**, 57, 9325–9329). In the repeated controlled potential electrolysis experiment, we optimized the electrochemical set-up, in which the auxiliary electrode was composed of a platinum wire wrapped with a copper wire instead of using ferrocene. We used a less negative reduction potential (−0.5 V) and a less positive oxidation potential (+0.7 V) to limit the degradation of the [3]catenane. Accordingly, the time for the oxidation step was extended from 10 to 15 min.

6. The description of the preparation of positionally biased isotopologues as “in a controlled manner” (p7, line 12-15) does not help the reader understand the nature of this experiment. A brief description of the synthetic strategy involving sequential threading would significantly improve readability of this section.

Reply: A brief description of the synthetic strategy involving sequential threading has been added to the revised manuscript.

7. The schematic and photograph of the electrolysis setup (Fig. 4a/b) does not aid understanding. There does not seem to be any significant innovation to this (if there is, it is not described; the same set up appears to have previously been reported in ref. 19). Including this detail in the main manuscript detracts from the novel concepts reported.

Reply: As suggested by the referee, the schematic and photograph of the electrolysis setup (Fig. 4a/b) have been removed from the main manuscript. Instead, they have been added to the revised Supplementary Information as Supplementary Fig.50.

8. p9, lines 6-9: "...dramatically different from previously reported [3]catenane-based molecular motors^{13,14} in that our electric molecular motor can operate continuously and synchronously, and is driven by a single energy source."

Achieving continuous and synchronous motion in one system is certainly a novel feature of the current example. However, the previously reported examples each also operate by "a single energy source" (photons in ref. 13 and protons in ref. 14). The differentiating factor is that other stimuli are required to operate these earlier machines. It should be made clear that these reagents do not provide energy that powers the motion.

Reply: The sentence has been changed to "This mechanism is different from previously reported [3]catenane-based molecular motors^{4,5} in that our electric molecular motor is driven by a single stimulus, namely the continuous oscillation of redox potential, i.e., an applied direct current voltage."

9. The mass spectrometry results (Suppl. Info. Section 4) require some interpretive explanation.

Reply: In order to assist the readers' understanding of the mass spectrum and ion-mobility mass spectrum, we have modified the Figure captions for Supplementary Fig. 33 and Supplementary Fig. 34 in the revised Supplementary Information.

10. There are some subtle but significant differences in the ¹H NMR spectra presented in Suppl. Fig. 35 (e.g. chemical shifts of the highlighted BPM protons). Similarly for the spectra in Suppl. Fig. 45. The authors should briefly comment on the origin of these differences.

Reply: On account of the different water contents in CD₃CN, there are subtle differences in the ¹H NMR spectra presented in Supplementary Fig. 35 and Fig. 45. Another reason for the

differences in the ^1H NMR spectra presented in Supplementary Fig. 35 is that the tested sample has not been purified after one redox cycle. It includes Zn^{2+} cations in the solution after the redox cycle. We have added these comments to the caption of the related Supplementary Fig. 35 and Fig. 51 in the revised Supplementary Information.

11. The significance or contribution of the data on the relaxation of the deuterated isotopologues presented in Suppl. Figs. 67 and 68 is not clear. This should be described in the Suppl. Info. or else these experiments removed.

Reply: These two figures have been removed in the revised Supplementary Information.

Reviewer Reports on the First Revision:

Referees' comments:

Referee #1 (Remarks to the Author):

I only have a very basic understanding of mechanics and wikipedia to guide me (<https://en.wikipedia.org/wiki/Gear>) but I don't agree with the authors that their system utilizes an intramolecular gearing mechanism. Their explanation (point 6 and their response to reviewer 3 on the same issue) has not persuaded me otherwise. Intermeshed gears do not rotate about the same axle as argued by the authors here. Personally, I find the analogy misleading rather than helpful.

Dietz and co-workers have recently published an electrically-driven rotary motor using DNA origami that has clear parallels to this work (functioning by an energy ratchet driven by the inherent oscillations of an alternating current). The authors could consider including that very recent reference: Pumm, A.-K. et al. A DNA origami rotary ratchet motor. Nature 607, 492–498 (2022).

Referee #2 (Remarks to the Author):

Thank you very much for polite revision cover letter. I satisfied the revised manuscript submitted by the authors. I appreciate that the authors also have interest in potential availability of molecular motor on record history of the number of rotations in a statistical manner. At the same time, I understand this is experimentally difficult at present. In the future, I hope this filed will develop in this regard.

Referee #3 (Remarks to the Author):

The authors have made numerous and substantial changes to their manuscript in response to the reviewers' comments. In doing so, they have addressed the majority of points raised by each reviewer and, in my opinion, have improved their manuscript. There are, however, a small number of remaining and arising points that I would still like to highlight.

1. Suppl. Info Section 13: Metastable State and Kinetic Studies

At the end of their expanded description of the kinetic studies, the authors make a statement, which I do not know how to interpret.

pS96 "In an analogous experiment in which the same sample after re-oxidation by NOPF6 was shaken several times before measurement, the kinetics was so fast that we were unable to identify any metastable species by ¹H NMR spectroscopy, i.e., the co-conformational rearrangement completed within a few seconds."

If I have understood this correctly, it suggests that the kinetics measured in the original experiment (over a period of ca. 1 h) are affected by the method of mixing, which must mean that the phenomena observed are a consequence of the rate of oxidation, not a direct measurement of the intramolecular co-conformational motion. This statement needs to be clarified.

2. Calculation of directionality of motion

I still find the discussion regarding estimating the directionality of motion poorly described. I understand from their rebuttal that the authors do not wish to “assume $\text{pred} = 1$ ”. However, this caution is set alongside statements in the main text emphasising the fidelity of the IPP barrier under reducing conditions:

e.g. p4 lines 23–25

“The height of the steric barrier imposed by the IPP unit is independent of the redox state and is large enough ($\Delta E > 25 \text{ kcal mol}^{-1}$) to preclude (Supplementary Figs. 37 and 38) the passage of the CBPQT4+/2(+•) ring over the IPP unit.”

And p7, line 25:

“As the CBPQT2(+•) ring does not pass over the IPP unit under reducing conditions”

Even in their discussion on pS94 it is stated “ $1 - \text{pred} \approx 0$ ” (consistent with my understanding above), but then “...so we set... $0.5 < \text{pred} \leq 1$ ” is mathematically inconsistent with the previous statement.

The authors should at least add a note to pS94 along the same lines as in their rebuttal to explain a cautionary approach, and even better, why 0.5 is an appropriate lower bound for pred .

In the end, their analysis leads to the same conclusion (directionality is ca. 85%) as the trivial conclusion based on setting $1 - \text{pred} = 0$, and consequently, $\text{pox} = 0.85$, which calls into question the added insight that the more convoluted analysis brings.

3. Relative rates of electron transfer processes and submolecular motions

At various places, all three referees raised points relating to the relative rates of each electron transfer event; the effect of CBPQT position on these processes; and the rates of electron transfer versus co-conformational motion. There is sufficient evidence that electron transfer is rapid in comparison to sub-molecular motion and, consequently, any co-conformational effect on electron-transfer rate state is not important; likewise, transfer to each of the two macrocycles and two redox-active units in the track are sufficiently fast they can be assumed to be “complete” before sub-molecular motion occurs (under the experimental conditions chosen).

However, the fact that these questions were posed by all three referees suggests that these assumptions and their experimental basis are not sufficiently emphasised. For clarity, I suggest that the authors should explicitly state these assumptions at an appropriate place in the discussion.

4. Choice of voltages and conditions in controlled potential electrolysis procedure

In their rebuttal, the authors nicely clarified their rationale for the choice of voltages and reaction times in the initial and repeated CPE experiments. However, the description in the manuscript is still a little opaque. For example, I cannot find a statement that the voltages were set based on the CV studies to achieve full oxidation and reduction and was furthermore informed by considerations of side reactions.

e.g. Supplementary Information pS75 “In a [sic] optimized¹⁸ reaction set-up (Supplementary Fig. 50), ...” does not explain the considerations regarding degradation in choosing the voltages and reaction times.

I appreciate this may seem obvious, but given the generalist target audience, I feel this would benefit understanding of the experimental rationale.

5. Analogic description of the molecular mechanism as involving “gearing interactions”

In spite of the rationale presented in their rebuttal, I still cannot agree that the molecular motor presented is usefully described as operating through “gearing interactions” and I note that referee

1 had similar reservations regarding this terminology.

My objection to this analogy does not stem from the “softer” nature of molecular motion compared to macroscopic mechanical devices. Rather, I do not find the essential elements of a gearing mechanism in the current system.

In macroscopic mechanics, a gear converts motion along/around one degree of freedom to another (e.g. rotary motion around two orthogonal axes; or rotary motion to linear motion) or from one angular velocity to another. Crucially, one end of the geared chain is powered externally, while the motion of the coupled element(s) follows in a fashion defined by the gearing ratio (e.g. the rotary motion of a steering column drives the linear motion of the wheels in a car; the rotary motion applied to the pedals on a bicycle at one angular velocity drives the rotary motion of the wheels at a different velocity). It does not matter which end of the geared system is powered, the coupling remains the same (e.g. applying a force from outside the car to change the direction of wheels would rotate the steering column). Furthermore, if the geared coupling is broken, the powered motion would still persist, but the coupled motion would not (e.g. I could still turn the pedals on my bicycle in a unidirectional fashion even if the chain were to break).

There is no analogous conversion of degree of freedom or velocity in the current system. The two rings move along the same track (degree of freedom) at the same (overall) rate. The movement of one ring is not ‘powered’ while the other ring ‘follows’. If one ring is removed, the other no longer moves in a unidirectional fashion either. The gearing ‘ratio’ is not 1:1 by chance; rather, there is no way this mechanism could be adapted to behave in any other way. That is to say, the motions of the two rings are more than simply coupled, they are part of one and the same motor mechanism – neither would move in a unidirectional fashion on its own.

By contrast, the earlier examples of molecular gears developed by Iwamura, Mislow, and others, much better match the macroscopic definition. Although in these early systems, there was no powering mechanism, if motion around one axis were to be powered externally, this would generate movement around a different axis at a ratio defined by the intermeshed ‘teeth’ of the molecular structures.

The authors correctly describe the directional motion as an “emergent” consequence of the [3]catenane design. In doing so, they identify a key aspect in which their molecular system does not match a macroscopic gear. A macroscopic gearing mechanism plays no part in generating unidirectional motion – it simply converts the unidirectional motion of one component into unidirectional motion of another.

This discussion does not impinge on the scientific veracity or significance of the experimental work. However, in a field that has previously been hampered by terminology that draws inaccurate or misleading parallels with macroscopic phenomena, I believe such semantic issues are important.

Author Rebuttals to First Revision:

Listed below are the details of our responses to the referees' comments.

Reviewers' comments

Referee #1:

I only have a very basic understanding of mechanics and wikipedia to guide me (<https://en.wikipedia.org/wiki/Gear>) but I don't agree with the authors that their system utilizes an intramolecular gearing mechanism. Their explanation (point 6 and their response to reviewer 3 on the same issue) has not persuaded me otherwise. Intermeshed gears do not rotate about the same axle as argued by the authors here. Personally, I find the analogy misleading rather than helpful.

Reply: We have removed the term of gearing interactions in the revised manuscript and added a brief discussion of how noncovalent interactions play a similar role in the microscopic and nanoscopic worlds to that of gears in macroscopic machines. See Page 10.

Dietz and co-workers have recently published an electrically-driven rotary motor using DNA origami that has clear parallels to this work (functioning by an energy ratchet driven by the inherent oscillations of an alternating current). The authors could consider including that very recent reference: Pumm, A.-K. et al. A DNA origami rotary ratchet motor. *Nature* 607, 492–498 (2022).

Reply: The reference (Pumm, A.-K. et al. A DNA origami rotary ratchet motor. *Nature* 607, 492–498 (2022).) has been cited in the revised main text.

Referee #2:

Thank you very much for polite revision cover letter. I satisfied the revised manuscript submitted by the authors. I appreciate that the authors also have interest in potential availability of molecular motor on record history of the number of rotations in a statistical manner. At the same time, I understand this is experimentally difficult at present. In the future, I hope this filed will develop in this regard.

Referee #3:

The authors have made numerous and substantial changes to their manuscript in response to the reviewers' comments. In doing so, they have addressed the majority of points raised by each reviewer and, in my opinion, have improved their manuscript. There are, however, a small number of remaining and arising points that I would still like to highlight.

1. Suppl. Info Section 13: Metastable State and Kinetic Studies

At the end of their expanded description of the kinetic studies, the authors make a statement, which I do not know how to interpret.

pS96 “In an analogous experiment in which the same sample after re-oxidation by NOPF₆ was shaken several times before measurement, the kinetics was so fast that we were unable to identify any metastable species by ¹H NMR spectroscopy, i.e., the co-conformational rearrangement completed within a few seconds.”

If I have understood this correctly, it suggests that the kinetics measured in the original experiment (over a period of ca. 1 h) are affected by the method of mixing, which must mean that the phenomena observed are a consequence of the rate of oxidation, not a direct measurement of the intramolecular co-conformational motion. This statement needs to be clarified.

Reply: In order to avoid misleading, the statement “In an analogous experiment in which the same sample after re-oxidation by NOPF₆ was shaken several times before measurement, the kinetics was so fast that we were unable to identify any metastable species by ¹H NMR spectroscopy, i.e., the co-conformational rearrangement completed within a few seconds.” has been removed in the revised Supplementary Information.

2. Calculation of directionality of motion

I still find the discussion regarding estimating the directionality of motion poorly described. I understand from their rebuttal that the authors do not wish to “assume $\text{pred} = 1$ ”. However, this caution is set alongside statements in the main text emphasising the fidelity of the IPP barrier under reducing conditions:

e.g. p4 lines 23–25

“The height of the steric barrier imposed by the IPP unit is independent of the redox state and is large enough ($\Delta E > 25 \text{ kcal mol}^{-1}$) to preclude (Supplementary Figs. 37 and 38) the passage of the CBPQT4+/2(+•) ring over the IPP unit.”

And p7, line 25: “As the CBPQT2(+•) ring does not pass over the IPP unit under reducing conditions”

Even in their discussion on pS94 it is stated “ $1 - \text{pred} \approx 0$ ” (consistent with my understanding above), but then “...so we set... $0.5 < \text{pred} \leq 1$ ” is mathematically inconsistent with the previous statement.

The authors should at least add a note to pS94 along the same lines as in their rebuttal to explain a cautionary approach, and even better, why 0.5 is an appropriate lower bound for pred .

In the end, their analysis leads to the same conclusion (directionality is ca. 85%) as the trivial conclusion based on setting $1 - \text{pred} = 0$, and consequently, $\text{pox} = 0.85$, which calls into question the added insight that the more convoluted analysis brings.

Reply: In order to avoid creating confusion, we have assumed $\text{pred} \approx 1$ and rewritten the discussion in the Section 12, Measurement of Directionality, of Supplementary Information accordingly.

3. Relative rates of electron transfer processes and submolecular motions

At various places, all three referees raised points relating to the relative rates of each electron transfer event; the effect of CBPQT position on these processes; and the rates of electron transfer versus co-conformational motion. There is sufficient evidence that electron transfer is rapid in comparison to sub-molecular motion and, consequently, any co-conformational effect on electron-transfer rate state is not important; likewise, transfer to each of the two macrocycles and two redox-active units in the track are sufficiently fast they can be assumed to be “complete” before sub-molecular motion occurs (under the experimental conditions chosen).

However, the fact that these questions were posed by all three referees suggests that these assumptions and their experimental basis are not sufficiently emphasized. For clarity, I suggest that the authors should explicitly state these assumptions at an appropriate place in the discussion.

Reply: As suggested by the referee, a discussion of relative rates of electron transfer processes and ring motions has been added to the Methods Section of the main text.

4. Choice of voltages and conditions in controlled potential electrolysis procedure

In their rebuttal, the authors nicely clarified their rationale for the choice of voltages and reaction times in the initial and repeated CPE experiments. However, the description in the manuscript is still a little opaque. For example, I cannot find a statement that the voltages were set based on the CV studies to achieve full oxidation and reduction and was furthermore informed by considerations of side reactions.

e.g. Supplementary Information pS75 “In a [sic] optimized¹⁸ reaction set-up (Supplementary Fig. 50), ...” does not explain the considerations regarding degradation in choosing the voltages and reaction times.

I appreciate this may seem obvious, but given the generalist target audience, I feel this would benefit understanding of the experimental rationale.

Reply: We have added a comment on the choice of voltages and reaction times used for the controlled potential electrolysis procedure in the revised Supplementary Information.

5. Analogic description of the molecular mechanism as involving “gearing interactions”

In spite of the rationale presented in their rebuttal, I still cannot agree that the molecular motor presented is usefully described as operating through “gearing interactions” and I note that referee 1 had similar reservations regarding this terminology.

My objection to this analogy does not stem from the “softer” nature of molecular motion compared to macroscopic mechanical devices. Rather, I do not find the essential elements of a gearing mechanism in the current system.

In macroscopic mechanics, a gear converts motion along/around one degree of freedom to another (e.g. rotary motion around two orthogonal axes; or rotary motion to linear motion) or from one angular velocity to another. Crucially, one end of the geared chain is powered externally, while the motion of the coupled element(s) follows in a fashion defined by the

gearing ratio (e.g. the rotary motion of a steering column drives the linear motion of the wheels in a car; the rotary motion applied to the pedals on a bicycle at one angular velocity drives the rotary motion of the wheels at a different velocity). It does not matter which end of the geared system is powered, the coupling remains the same (e.g. applying a force from outside the car to change the direction of wheels would rotate the steering column). Furthermore, if the geared coupling is broken, the powered motion would still persist, but the coupled motion would not (e.g. I could still turn the pedals on my bicycle in a unidirectional fashion even if the chain were to break).

There is no analogous conversion of degree of freedom or velocity in the current system. The two rings move along the same track (degree of freedom) at the same (overall) rate. The movement of one ring is not ‘powered’ while the other ring ‘follows’. If one ring is removed, the other no longer moves in a unidirectional fashion either. The gearing ‘ratio’ is not 1:1 by chance; rather, there is no way this mechanism could be adapted to behave in any other way. That is to say, the motions of the two rings are more than simply coupled, they are part of one and the same motor mechanism – neither would move in a unidirectional fashion on its own.

By contrast, the earlier examples of molecular gears developed by Iwamura, Mislow, and others, much better match the macroscopic definition. Although in these early systems, there was no powering mechanism, if motion around one axis were to be powered externally, this would generate movement around a different axis at a ratio defined by the intermeshed ‘teeth’ of the molecular structures.

The authors correctly describe the directional motion as an “emergent” consequence of the [3]catenane design. In doing so, they identify a key aspect in which their molecular system does not match a macroscopic gear. A macroscopic gearing mechanism plays no part in generating unidirectional motion – it simply converts the unidirectional motion of one component into unidirectional motion of another.

This discussion does not impinge on the scientific veracity or significance of the experimental work. However, in a field that has previously been hampered by terminology that draws inaccurate or misleading parallels with macroscopic phenomena, I believe such semantic issues are important.

Reply: We have removed the term of gearing interactions in the revised manuscript and added a brief discussion of how noncovalent interactions play a similar role in the microscopic and nanoscopic worlds to that of gears in macroscopic machines. See Page 10.